# On physical mechanisms enhancing air–sea $CO_2$ exchange

Lucía Gutiérrez-Loza[1], Erik Nilsson[1], Marcus B. Wallin[1,2], Erik Sahlée[1], and Anna Rutgersson[1]

[1]Department of Earth Sciences, Uppsala University, Uppsala, Sweden
[2]Department of Aquatic Sciences and Assessment, Swedish University of agricultural Sciences, Uppsala, Sweden

**Correspondence:** Lucía Gutiérrez-Loza (lucia.gutierrez_loza@geo.uu.se)

**Abstract.**

Reducing uncertainties in the air–sea $CO_2$ flux calculations is one of the major challenges when addressing the oceanic contribution in the global carbon balance. In traditional models, the air–sea $CO_2$ flux is estimated using expressions of the gas transfer velocity as a function of wind speed. However, other mechanisms affecting the variability in the flux at local and regional scales are still poorly understood. The uncertainties associated with the flux estimates become particularly large in heterogeneous environments such as coastal and marginal seas. Here, we investigated the air–sea $CO_2$ exchange at a coastal site in the central Baltic Sea using nine years of eddy covariance measurements. Based on these observations we were able to capture the temporal variability of the air–sea $CO_2$ flux and other parameters relevant for the gas exchange. Our results show that a wind-based model with similar pattern to those developed for larger basins and open sea condition can, on average, be a good approximation for $k$. However, in order to reduce the uncertainty associated to these averages and produce reliable short-term $k$ estimates, additional physical processes must be considered. Using a normalized gas transfer velocity, we identified conditions associated to enhanced exchange (large $k$ values). During high and intermediate wind speeds (above 6–8 $\mathrm{m\,s^{-1}}$), conditions on both sides of the air–water interface were found to be relevant for the gas exchange. Our findings further suggest that at such relatively high wind speeds, sea spray is an efficient mechanisms for air–sea $CO_2$ exchange. During low wind speeds (<6 $\mathrm{m\,s^{-1}}$), water-side convection was found to be a relevant control mechanism. The effect of both sea spray and water-side convection on the gas exchange showed a clear seasonality with positive fluxes (winter conditions) being the most affected.

## 1 Introduction

Air–sea $CO_2$ exchange is an essential aspect of the global carbon cycle, having great implications for the Earth's climate. The global oceans are estimated to be net sinks of $CO_2$ taking up on average ca 25 % of the $CO_2$ emitted every year to the atmosphere due to anthropogenic activities (Friedlingstein et al., 2021). The current global ocean uptake is estimated to be between -2.0 and -3.1 $\mathrm{GtC\,yr^{-1}}$ (Takahashi et al., 2009; Ciais et al., 2013; Friedlingstein et al., 2021). However, large uncertainties are still associated with the air–sea $CO_2$ flux estimates, mainly due to the incomplete understanding of the spatio-temporal variability in the controlling mechanisms. Resolving the effect of these mechanisms at the relevant temporal and spatial scales is essential to constrain the oceanic contribution in the global carbon balance.

The exchange of $CO_2$ across the air–sea interface can be described using the following bulk formula:

$$FCO_2 = kK_0\Delta pCO_2 = kK_0(pCO_2^w - pCO_2^a),\tag{1}$$

where the air–sea $CO_2$ flux ($FCO_2$) is a function of the gas transfer velocity ($k$), the difference in the partial pressure of $CO_2$ ($\Delta pCO_2$) between the atmosphere and the seawater (superscripts $a$ and $w$, respectively), and the salinity- and temperature-
dependent solubility constant ($K_0$). The direction of the flux is determined by the sign of $\Delta pCO_2$ and, by convention, positive (upward) $FCO_2$ represents transport from the ocean to the atmosphere (i.e. positive $\Delta pCO_2$).

The gas transfer velocity, $k$, represents the efficiency of the transfer processes across the air–sea interface. For $CO_2$ and other slightly soluble gases, such efficiency is particularly associated to the turbulent processes occurring in the oceanic boundary layer, which ultimately control the air–sea gas exchange. The wind can be associated, directly or indirectly, with most of the
turbulent process near the ocean–atmosphere interface. Thus, the traditional approach suggests that $k$ can be represented as a function of the the wind speed since it is the largest source of kinetic energy to the upper ocean. With wind-speed data being a widely available resource globally, wind-based parametrizations of $k$ have often been used to obtain global estimates of $FCO_2$ (e.g. Takahashi et al., 2009). However, large uncertainties in the estimated $FCO_2$ have been associated to the uncertainties in $k$ (Woolf et al., 2019). At regional and local scales, the magnitude of these uncertainties becomes especially problematic,
particularly in coastal environments where adequate representation of the physical and biogeochemical processes, and their interactions, is necessary in order to avoid large biases in the flux estimates.

In addition to the wind speed, other water-side control mechanisms are well known to play a significant role in the gas transfer processes of slightly soluble gases, such as $CO_2$. Furthermore, the effect of atmospheric controls, and their impact on the upper layer of the ocean, are potentially relevant (e.g. Erickson III, 1993) but seldom considered. The relative importance
of these forcing mechanisms on the gas exchange is highly dependent on the characteristics near the sea surface, which in turn, can be categorized based on wind-speed regimes (e.g. Soloviev and Lukas, 2013). At moderately high wind speeds, above $8$–$10\,\mathrm{m\,s^{-1}}$, the upper layer of the ocean is generally well mixed (from the surface up to several tens of meters depth). Under these conditions, breaking waves (Zhao et al., 2003; Blomquist et al., 2017; Brumer et al., 2017), bubbles (Woolf, 1993, 1997; Bell et al., 2017), and sea spray (Andreas et al., 2016) have a significant role on air–sea interaction processes. At wind speeds
lower than $4$–$5\,\mathrm{m\,s^{-1}}$, when the effects of breaking waves and wind-induced mixing are limited, convective processes in the atmosphere (Erickson III, 1993) and the sea (Rutgersson and Smedman, 2010) become relevant. Other processes such as surface films (Pereira et al., 2018; Ribas-Ribas et al., 2018), rain (Ashton et al., 2016), Langmuir circulation (Thorpe et al., 2003), and micro-scale wave breaking (Jessup et al., 1997) might be relevant over a wider range of wind speeds, including intermediate wind velocities. In order to explain the variability and reduce the uncertainty in $FCO_2$ estimates, it is necessary
to understand the effect of the control mechanisms on gas exchange, particularly at higher wind speeds but even relevant at low and moderate wind speeds.

Coastal oceans and marginal seas are active and heterogeneous environments in terms of both physical and biogeochemical processes. These regions have been found to be net sinks of $CO_2$ at global scales (Borges et al., 2005; Laruelle et al., 2010; Chen et al., 2013) with disproportionately large contributions to the global carbon system when compared to the open ocean

(Laruelle et al., 2014). The complexity and heterogeneity of the coastal regions causes large spatio-temporal variability in the air–sea $CO_2$ exchange (Roobaert et al., 2019), variability that is rarely accounted for in global estimates.

The Baltic Sea is a semi-enclosed sea located at relatively high latitudes, stretching from $54°$ N to $66°$ N. The basin is largely affected by terrestrial inputs from surrounding watersheds and has relatively limited water exchange with the open ocean. This leads to a dynamic carbon system with significant spatio-temporal variability. A thorough assessment of the biogeochemical functioning of the Baltic Sea was recently published by Kuliński et al. (2021). In terms of the air–sea $CO_2$ exchange, several approaches have been used to estimate the regional fluxes, however, no consensus has been reached on the role of the Baltic Sea as a net source or sink of atmospheric $CO_2$ (Thomas et al., 2010; Kuliński and Pempkowiak, 2011; Norman et al., 2013b; Parard et al., 2017). In order to resolve some of the key elements associated to the air–sea $CO_2$ exchange, previous studies have focused on the diurnal (Honkanen et al., 2021) and seasonal (Thomas and Schneider, 1999b; Rutgersson et al., 2008; Schneider et al., 2014) variability in the partial pressure of $CO_2$ across the Baltic Sea, and on the spatial and temporal variability in the atmospheric $CO_2$ concentrations (Rutgersson et al., 2009). Furthermore, water-side convection (Rutgersson and Smedman, 2010; Norman et al., 2013b), upwelling events (Norman et al., 2013a; Jacobs et al., 2021), and ice coverage (Löffler et al., 2012) have all been recognized as important regional controls on the gas exchange. Despite these efforts, the effect of the different mechanisms modulating the air–sea gas exchange, and its variability, is still poorly understood in the Baltic Sea, as it is in many other coastal regions. Limited data availability is the main reason hindering our ability to resolve processes at relevant spatial and temporal scales. In this context, continuous and long-term monitoring of the air–sea $CO_2$ exchange in coastal areas is essential to improve our understanding of the gas transfer mechanisms.

In this study, we present and evaluate data collected during a nine-year period at the land-based station Östergarnsholm located on an island in the central Baltic Sea. This is, to the best of our knowledge, the longest record of air–sea $CO_2$ flux based on eddy covariance measurements. Using atmospheric and water-side data we evaluated different control mechanisms modulating the gas transfer velocity, $k$, covering a wide range of wind speed conditions.

## 2 Site and Data

### 2.1 The Östergarnsholm site

We used data collected between 2013 and 2021 from the Swedish marine Integrated Carbon Observation System (ICOS) station, Östergarnsholm. The station ($57°$ 27' N, $18°$ 59' E) is located on a small and flat island located $4\,\mathrm{km}$ east of the bigger island of Gotland in the central Baltic Sea (Fig. 1). Measurements are performed in a $30\,\mathrm{m}$ land-based tower located on the southern tip of the island, with the base of the tower at $1.4\,\mathrm{m}$ above the mean sea level (Sjöblom and Smedman, 2002). The tower has been used to monitor and study the marine atmospheric boundary layer and air–sea interaction processes since 1995 (e.g. Smedman et al., 1999; Rutgersson et al., 2001; Högström et al., 2008; Rutgersson and Smedman, 2010; Rutgersson et al., 2020).

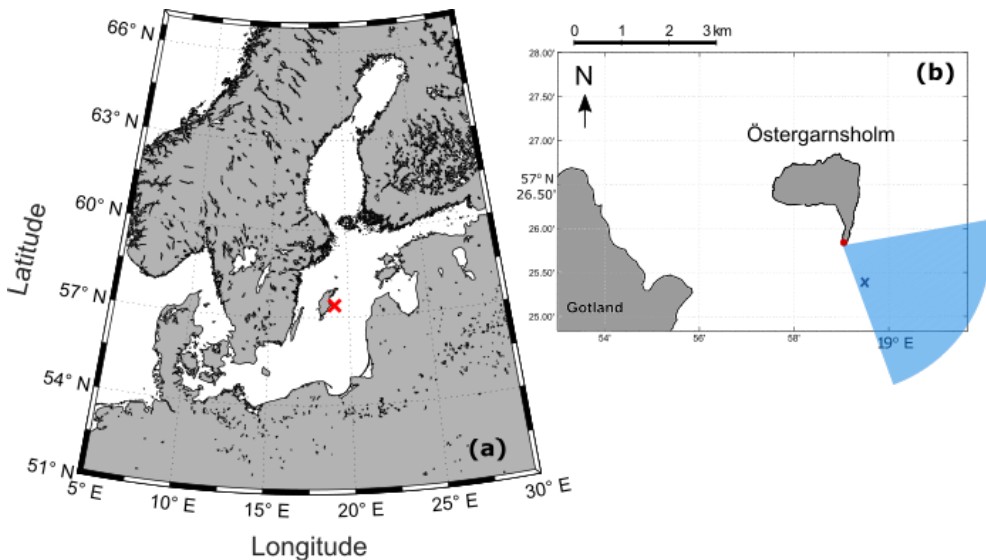

**Figure 1.** (a) Map of the Baltic Sea; the red cross in the central Baltic Sea indicates the location of the Östergarnsholm station. (b) Map of the Östergarnsholm station ca 4 km off from the Gotland Island; the red dot indicates the location of the tower, the blue cross is the location of the mooring with water-side instrumentation (Sect. 2.1.2), and the shaded blue area is the so-called "open-sea" sector with wind directions from $80° < \text{WD} < 160°$ (see Sect. 2.1.1 for details).

### 2.1.1 Atmospheric data

The tower at Östergarnsholm was instrumented with high-frequency sensors for continuous turbulence measurements. Air–sea $CO_2$ fluxes were calculated using the fluctuations of the vertical wind component measured by a CSAT3-3D sonic anemometer (Campbell Scientific, Inc., Logan, UT, USA) and the fluctuations of the atmospheric $CO_2$ molar densities measured with an open-path gas analyzer model LI-7500A (from 2013 to 2017) and model LI-7500RS (between 2017 and 2021) (LI-COR, Inc., Lincoln, NE, USA). For details about the flux calculations see Sect. 2.2. The sonic anemometer and gas analyzer were located at a height of 9 m from the tower base (i.e. 10.4 m with respect to the mean sea level), and had a sampling rate of 20 Hz. In addition to the high-frequency data, profile measurements of wind speed, wind direction, and temperature at 7, 12, 14, 20, and 29 m height were carried out at 1 Hz and averaged over 30 min periods. Relative humidity, atmospheric pressure, incoming solar radiation, and precipitation were also measured at the site.

The measurements at the Östergarnsholm site have been found to be representative of open-sea or coastal conditions depending on the wind direction (Rutgersson et al., 2020). Only data with wind directions from southeast ($80° < \text{WD} < 160°$) representing open-sea conditions were included in the analysis of the current study (Fig. 1b). For this wind sector, it was considered that no disturbances occurred in the tower measurements due to flow distortion, and that the wave field was not affected by the shallowing of the seafloor (Högström et al., 2008). Furthermore, the biogeochemical water properties and hydrographycal features were assumed to be spatially homogeneous along this sector (Rutgersson et al., 2008, 2020), ensuring

that the water-side measurements were representative of the flux footprint of the tower. A more detailed description of the Östergarnsholm site can be found in Rutgersson et al. (2020).

The flux footprint is a function used to characterize the contributions of the sources and sinks per unit area to the total flux measured at a certain point. Based on this mathematical concept, it is possible to associate the fluxes measured at a specific height with the surface exchange of any scalar (e.g. Kormann and Meixner, 2001). According to the flux footprint estimates (Kljun et al., 2015), in this study the source/sink area of the fluxes from the open-sea sector measured at 10.4 m height was located a few hundred meters upwind from the tower. Figure 2 shows the spatial distribution of the flux contributions (in $m^{-2}$) for different atmospheric stability conditions. For unstable and neutral conditions, the main flux source/sink was found as a localized area near the tower. While for stable conditions, the source/sink contributions per unit area were smaller closer to the tower and rather spread over a larger region.

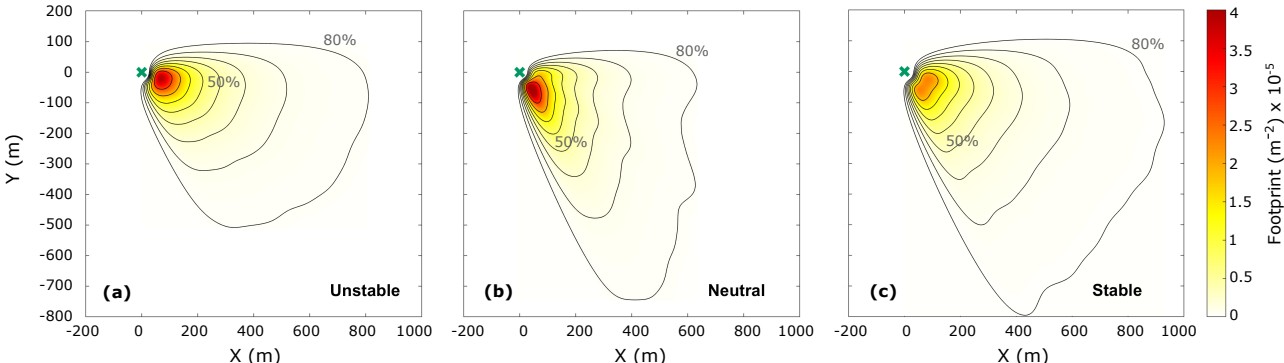

**Figure 2.** Average footprint distribution for (a) unstable, (b) neutral, and (c) stable atmospheric conditions. $X$ and $Y$ represent the horizontal domain with origin (0,0) at the tower's location (green cross), and the contours represent the percentage of source area from 10–80 %. The flux footprint (in color) shows the spatial distribution of the contributions per unit area to the total $FCO_2$ (in $m^{-2}$). The footprint was calculated using the model developed by Kljun et al. (2015) using all data available for the open-sea sector between mid-2013 and 2020.

The atmospheric stability was represented as $z/L$, where $z$ is the measurement height and $L$ is the Monin-Obukhov length. The latter is given by $L = -u_*^3 \Theta / g\kappa \overline{w'T_s'}$, where $u_*$ is the friction velocity, $\Theta$ is the potential temperature, $g = 9.81 \ \mathrm{m\,s^{-2}}$ is the acceleration due to gravity, $\kappa = 0.4$ is the Von Karman constant, and $\overline{w'T_s'}$ is the buoyancy flux. The sonic temperature, $T_s$, was considered to be almost equal to the virtual temperature, $T_v$, which is often used for buoyancy flux calculations (Aubinet et al., 2012). Following Sjöblom and Smedman (2002), we use $z = 10.4$ m as the mean measurement height with respect to the mean sea level. Here, unstable conditions were defined as $z/L \leq$ -0.05, near-neutral conditions -0.05 $< z/L <$ 0.05, and stable conditions $z/L \geq 0.05$.

### 2.1.2 Water-side measurements

Water-side measurements were carried out continuously at a mooring located $1\,\mathrm{km}$ southeast of the tower (see Fig.1b). At the mooring, seawater temperature and partial pressure of $CO_2$ were measured every $30\,\mathrm{min}$ using a SAMI-$CO_2$ sensor (Sunburst Sensors, LLC, MT, USA) at a depth of $4\,\mathrm{m}$. Additionally, continuous wave measurements were made with a Directional Waverider buoy located $4\,\mathrm{km}$ southeast of the tower (outside the domain in Fig. 1b). The Waverider buoy is operated and maintained by the Finnish Meteorological Institute (FMI). In this study, the characteristics of the sea state were represented by the significant wave height ($H_s$), the wave steepness calculated as $H_s/L_p$ were $L_p$ is the peak wavelength, and the wave age, $C_p/U_{10N}$, were $C_p$ is the phase velocity of the waves and $U_{10N}$ is the neutral equivalent wind speed at $10\,\mathrm{m}$ height.

Daily data of the mixed layer depth (MLD) from the Baltic Sea Reanalysis product provided by the Copernicus Marine Environment Monitoring System (CMEMS) (Von Schuckmann et al., 2016) were used as an indicator of the vertical mixing in the water column, and for the water-side convection calculations (Rutgersson and Smedman, 2010; Norman et al., 2013b). These data are freely available from CMEMS website at http://marine.copernicus.eu/services-portfolio/access-to-products/.

## 2.2 Data processing

High frequency data obtained at Östergarnsholm site (Sect. 2.1.1) were used for $FCO_2$ calculations using the eddy covariance method (Baldocchi et al., 1988; Aubinet et al., 2012) and the subsequent gas transfer velocity analysis. The turbulent fluctuations for the flux calculation (described below) were obtained following a Reynolds decomposition using an average period of $10\,\mathrm{min}$. These averages were calculated as block averages from the linearly detrended time series of the $20\,\mathrm{Hz}$ data. The turbulent fluctuations were used to calculate variances and covariances. Other statistical moments were also calculated from the turbulent fluctuations and used as part of the quality control.

The raw wind-speed components were transformed to earth-system coordinates and corrected using a double rotation (Kaimal and Finnigan, 1994). Wind speed and direction were calculated from the corrected components to avoid effects caused by the tilting of the sonic anemometer. Following the convention, measured wind speeds were adjusted to a neutral equivalent wind speed at $10\,\mathrm{m}$ height ($U_{10N}$). Only wind directions representing open-sea conditions were used in the analysis; see Sect. 2.1.1 for details. Only periods with the three consecutive $10\,\mathrm{min}$ averages from $80\,^\circ < \mathrm{WD} < 160\,^\circ$ (i.e. open-sea sector) were included. Data with wind speeds lower than $2\,\mathrm{m\,s^{-1}}$ were excluded of the $FCO_2$ calculations.

The performance of the gas analyzer was evaluated based on the relative signal strength indicator (RSSI). Following Nilsson et al. (2018), we used the variance of the RSSI ($\sigma^2_{RSSI}$) to remove low quality data. The variance was calculated over the $10\,\mathrm{min}$ periods, and only data with $\sigma^2_{RSSI} < 0.001$ were considered in the analysis. Additionally, thresholds of different statistical parameters were used to ensure the homogeneity of the data and avoid outliers. Data was excluded if the absolute value of the fourth order moment of the $CO_2$ signal was higher than $100\,\mathrm{ppm^4}$ to filter out outliers, and the variance of the vertical wind speed was $\sigma^2_w < 1e^{-6}\,\mathrm{m^2\,s^{-2}}$ to exclude unrealistically low values of the vertical wind variance (low turbulence conditions).

According to the eddy covariance method, the $FCO_2$ were calculated from:

$$FCO_2 = \rho_a \overline{w'c'}, \tag{2}$$

where, $\rho_a$ is the mean density of dry air, and the term $\overline{w'c'}$ represents the time-averaged covariance between the turbulent fluctuations of the vertical wind component ($w$) and the dry mole fraction of the gas ($c$). The $FCO_2$ were calculated over 30 min periods by averaging three consecutive 10 min periods fulfilling all the quality-control steps. The flux was directly calculated from the $CO_2$ dry mole fraction (i.e. mole fraction of $CO_2$ relative to dry air) obtained from the measured molar densities of $CO_2$ relative to the ambient air. By using this direct conversion method (Sahlée et al., 2008), the corrections for dilution effects (Webb et al., 1980) were avoided. A description of the direct conversion method and detailed discussion can be found in Sahlée et al. (2008). Fluxes with magnitudes below a minimum detection limit of $\pm 0.05\,\mu\mathrm{mol\,m^{-2}\,s^{-1}}$ were removed. This limit was empirically defined to avoid data with low signal-to-noise ratio. In addition to $FCO_2$, enthalpy fluxes were also estimated from the turbulent measurements as the sum of the sensible ($\rho C_p \overline{w'T'}$) and latent ($\rho \lambda \overline{w'q'}$) heat fluxes. Furthermore, cases with high relative humidity conditions (RH$> 95\,\%$) were excluded to avoid data possibly affected by condensation on the instruments.

The gas transfer velocity, $k$, was calculated from Eq. 1 using the calculated $FCO_2$ (Eq. 2). The solubility constant ($K_0$) was determined from the relationship suggested by Weiss (1974) using a constant salinity value of 7 PSU and in situ water temperature from the SAMI sensor. Changes in the salinity, which oscillates between 6.5 and 7.5 PSU in this region of the Baltic (e.g. Wesslander et al., 2010; Rutgersson et al., 2020), are not expected to have significant effects on the solubility. The $\Delta pCO_2$ was obtained from $pCO_2^w$ measured with the SAMI sensor, and $pCO_2^a$ calculated from the molar densities obtained with the gas analyzer. Periods with $\Delta pCO_2 < \pm 50\,\mu\mathrm{atm}$ were excluded from the analysis. Furthermore, during conditions of strong water-side stratification, $pCO_2^w$ measurements carried out at 4 m depth might not be representative of the air–sea $CO_2$ fluxes measured at the tower. Therefore, all the data occurring during strongly stratified conditions according to the in situ observations were not considered in the analysis. Data were removed when the water-side temperature gradient ($\Delta T_w$) was larger than $1\,^{\circ}\mathrm{C}$. The $\Delta T_w$ was defined as the difference between $T_w$ measured at 4 m depth and the near-surface water temperature ($T_{ns}$) measured at 0.35 m depth with the Waverider buoy. The calculated $k$ values were adjusted to a reference Schmidt number ($Sc$) to compensate for temperature and salinity effects. The adjusted gas transfer velocity ($k_{660}$) was given by $k_{660} = k(660/Sc)^{-1/2}$, where $Sc = 660$ corresponds to the Schmidt number of $CO_2$ at $20\,^{\circ}\mathrm{C}$ for seawater (S = $35\,\permil$), and $Sc$ was the Schmidt number calculated with the corresponding in situ water temperature ($T_w$) for each data point.

The final data set, after quality control processing, consisted of 3,477 $FCO_2$ data points and 1,349 $k_{660}$ data points. This amount of data corresponds to $18.7\,\%$ and $15\,\%$, respectively, out of the total amount of data available for the open-sea sector (18,625 $FCO_2$ and 8,974 $k_{660}$ data points). Further information about the rejection rates of each quality control criterion can be found in Appendix B.

The calculated $k_{660}$ were used to study the effect of water-side and atmospheric control mechanisms on air–sea $CO_2$ exchange. A wind-speed relationship ($k_{wind}$) was calculated as the cubic (best) fit to the bin-averaged $k_{660}$, using equidensity bins based on the wind speed percentiles, and used to obtain a normalized gas transfer velocity defined as $k_{660}/k_{wind}$. The use

of a normalized gas transfer velocity allowed the analysis of mechanisms on $k_{660}$, in addition to the effect of wind speed. For the analysis, the data were divided into three different wind-speed regimes from light breeze to moderate gale. The thresholds for these regimes were chosen depending on the expected conditions at the sea surface according to the Beaufort Scale (Barua, 2005). Low wind speeds were defined as $U_{10N} < 6\,\mathrm{m\,s^{-1}}$, covering conditions of light to moderate breeze; only ripples and small waves causing little disturbance on the surface were expected under these conditions. Intermediate conditions representing moderate breezes included wind speeds of $6 < U_{10N} < 8\,\mathrm{m\,s^{-1}}$. Finally, relatively high wind-speed conditions were defined as $U_{10N} > 8\,\mathrm{m\,s^{-1}}$ when moderate to long waves were expected, whitecaps and sea spray were likely to be observed; these wind speeds correspond to fresh breeze to moderate gale.

## 3   Results

### 3.1   Oceanic and meteorological conditions: the annual cycle

The annual cycle, obtained using the nine years of data (2013–2021), showed a seasonal pattern in both $pCO_2^w$ and $pCO_2^a$ (Fig. 3a). The $pCO_2^a$ variability was small, at least when compared to the variability in $pCO_2^w$. The lowest $pCO_2^a$ were observed during the late summer and autumn with values often below $380\,\mathrm{\mu atm}$. Higher values occurred during winter, reaching $440\,\mathrm{\mu atm}$. While the monthly means of $pCO_2^a$ oscillated around $410\,\mathrm{\mu atm}$. An increasing trend in $pCO_2^a$ was observed during the study period (not shown); a linear regression using the monthly averages suggested an increase of $0.2\,\mathrm{ppm}$ per month which corresponds to a total increase of approximately $20\,\mathrm{ppm}$ during the nine-year period. Both the trend and the seasonal variability in $pCO_2^a$ were masked by the variability in $pCO_2^w$ (Fig.3a). A strong seasonal pattern was observed for $pCO_2^w$ with values higher than those in the atmosphere during the winter, and lower during summer. The seasonality in $pCO_2^w$ in the Baltic Sea has been recognized to be strongly modulated by the biological activity (Thomas and Schneider, 1999a; Wesslander et al., 2010). Here, the lowest $pCO_2^w$ reached values below $100\,\mathrm{\mu atm}$ during the summer of 2018. The highest values of $pCO_2^w$ occurred during the winters of 2018–2019 and 2019–2020 with observed values higher than $800\,\mathrm{\mu atm}$. Furthermore, lower summer $pCO_2^w$ values were observed in the last three years in comparison to previous years. In this way, the inter-annual variability of $pCO_2^w$ was mostly noticeable in terms of an increasing amplitude of the seasonal cycle during the last years.

The monthly means of $FCO_2$ showed a seasonal cycle with positive fluxes during the winter and negative during the summer (Fig. 3b). This seasonal pattern in the flux was consistent with the thermodynamic forcing (i.e. $\Delta pCO_2 = pCO_2^w - pCO_2^a$) which suggested an upward transport during the winter and downward during the summer. However, a high variability in the half-hourly data was observed year-round.

The atmospheric and water-side variables describing the physical characteristics showed clear seasonal cycles (Fig. 4). However, a large scatter was observed from the individual half-hourly values, thus, highlighting the large variability and heterogeneity of the environment. The monthly means of wind speed and significant wave height ($H_s$) were higher during the autumn and winter, in comparison to the summer (Fig. 4a and 4b). However, short-term events with high winds ($U_{10N} > 10\,\mathrm{m\,s^{-1}}$) and waves ($H_s \sim 3\,\mathrm{m}$) were observed during all seasons. The temperature gradient ($\Delta T = T_w - T_a$, Fig. 4c) showed that from September to February (DOY 250–50) the ocean was, on average, $1.5\,^\circ\mathrm{C}$ warmer than the overlying air (i.e. positive gradient).

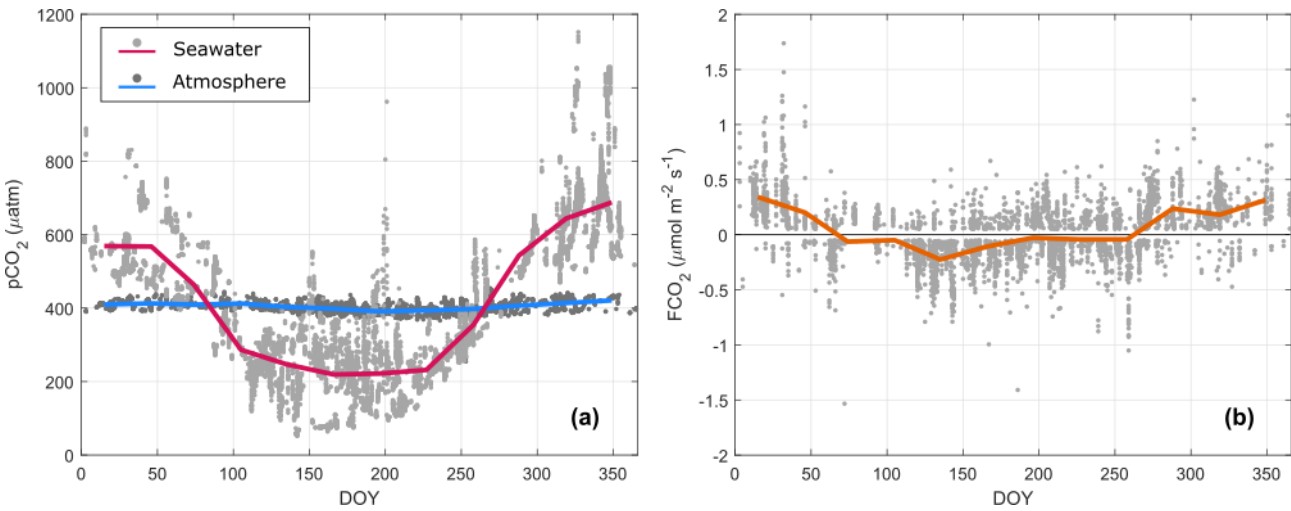

**Figure 3.** Annual cycle of (a) $CO_2$ partial pressure ($pCO_2$) in the seawater and in the atmosphere, and (b) air–sea $CO_2$ fluxes from eddy covariance. The dots represent the half-hourly values while the solid lines show the monthly averages.

During late spring, the atmospheric temperature was warmer than the seawater with an average $\Delta T$ of -1 °C. The enthalpy flux showed mean monthly values of 0–75 W m$^{-2}$ (Fig. 4d). The monthly means of relative humidity (RH, Fig. 4e), ranged between
225    60 and 85 % throughout the year, but a large scatter was observed. Particularly during autumn and winter (DOY 250–50), when moderately low values of RH (<40 %) often occurred.

### 3.2    The gas transfer velocity

The gas transfer velocity, $k_{660}$, followed—on average—an increasing relationship with wind speed. The best fit calculated using the mean $k_{660}$ values for each equidensity bin (130 data points) indicated a cubic relationship with wind speed, following
230    a general agreement with other commonly-used parameterizations (Fig. 5). Even when the best fit to the bin means ($k_{wind}$) seemed to accurately represent the average behaviour of $k_{660}$ as a function of wind speed ($R^2$ = 0.97), a large scatter in the $k_{660}$ half-hourly values was observed. Only $\sim 30$ % of such variability in the half-hourly values was explained by a wind speed relationship (not shown).

We used the gas transfer velocity normalized by the wind-based relationship (i.e. $k_{660}/k_{wind}$) to evaluate the effect of
235    individual atmospheric and water-side processes on the gas exchange. Boxplots showing the statistical summary for equidensity bins defined based on the distribution of the normalized gas transfer velocity as a function of each of the parameters were used to evaluate $k_{660}/k_{wind}$. Based on this analysis, we further identified the conditions inducing the strongest variability in $k_{660}$ under different wind speed regimes (Sect. 2.2) presented in Sect. 3.2.1 to 3.2.3. The entire set of figures of $k_{660}/k_{wind}$ can be found in Appendix A.

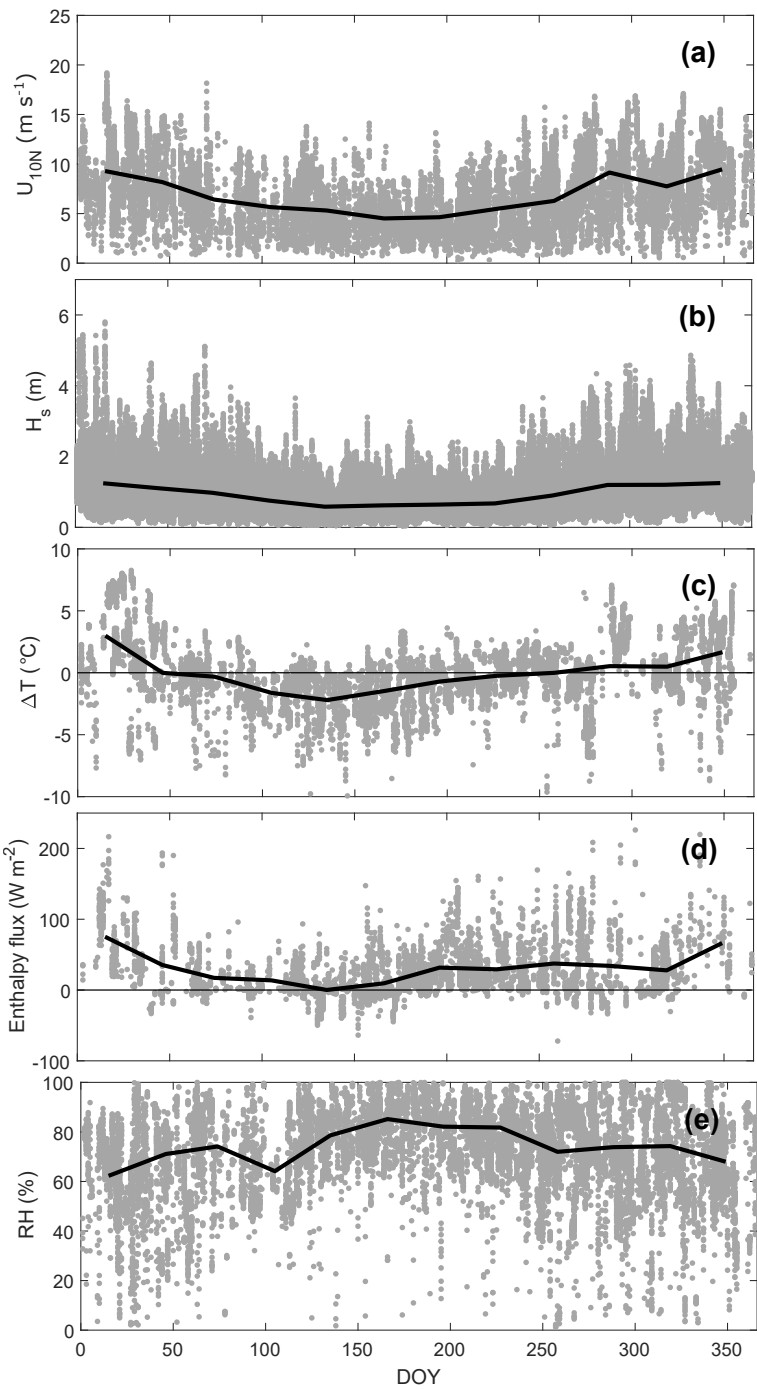

**Figure 4.** Annual cycle of (a) 10 m neutral wind speed, (b) significant wave height, (c) temperature gradient ($\Delta T = T_w - T_a$), (d) enthalpy flux, and (e) relative humidity. The dots are the half-hourly values while the solid black lines represent the monthly average of each parameter.

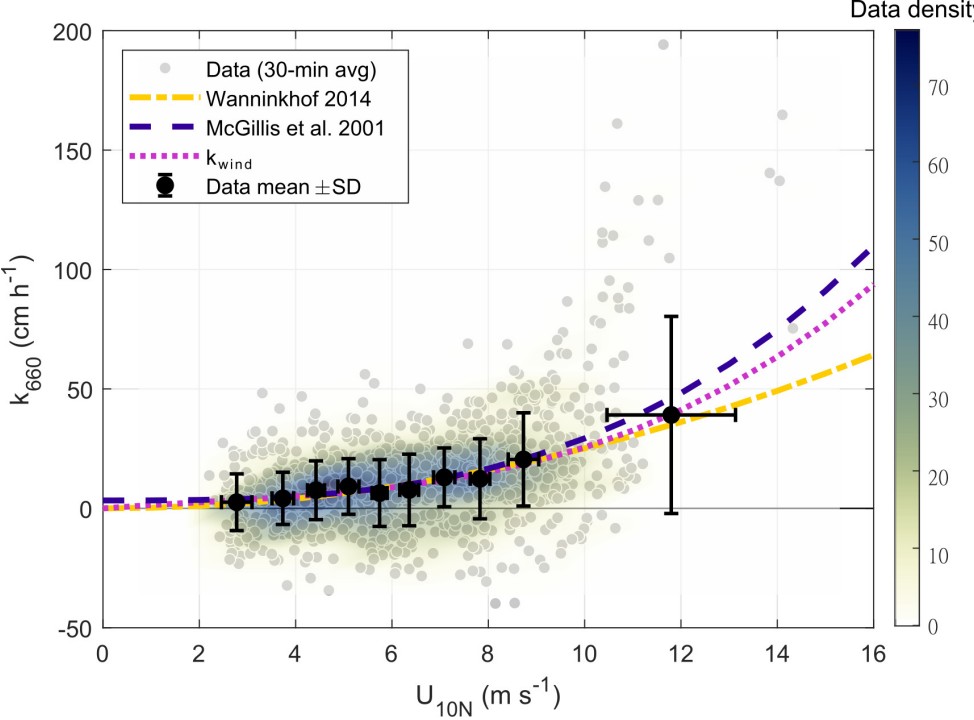

**Figure 5.** Gas transfer velocity for $CO_2$ (adjusted to a Schmidt number of 660) as a function of the 10 m neutral wind speed. The grey dots represent the half-hourly values of $k_{660}$ for the nine-year period from 2013 to 2021. The black dots and bars, represent the $k_{660}$ mean values and standard deviations, respectively, calculated for equi-density bins based on the wind speed percentiles; the best fit to the means is shown as the pink dotted line ($k_{wind}$). For reference, a quadratic (Wanninkhof, 2014) and cubic (McGillis et al., 2001) wind-based parametrizations were included. The colors in the shaded area represent the data density (in counts) with a grid bin size of $1\,\mathrm{m\,s^{-1}}$ by $10\,\mathrm{cm\,h^{-1}}$.

### 3.2.1 Controls on $k_{660}$ at high wind speed conditions

At high wind speed conditions ($U_{10N} > 8\,\mathrm{m\,s^{-1}}$), water-side properties such MLD and the wave field were found to be associated to the behaviour of the gas transfer velocity. Furthermore, $\Delta pCO_2$ was not only the driver of the flux defining the direction of the transport, but the characteristics of the gradient were also connected to the efficiency of the exchange. Thus, $\Delta pCO_2$ was considered here as a water-side control due to the importance of $pCO_2^w$ in modulating the variability of the gradient, in comparison to the rather constant values of $pCO_2^a$ (see Sect. 3.1).

Particular conditions of the water-side parameters, were found to be linked to $k_{660}$ values higher than those suggested by the wind-speed relationship (i.e. $k_{660}/k_{wind} > 1$), while the rest of the time, wind speed seemed to better describe the behaviour of $k_{660}$ (i.e. $k_{660}/k_{wind} \sim 1$) (Fig. 6). Moderately positive gradients, with $\Delta pCO_2$ of the order of 50-100 $\mu$atm, were associated with significantly higher $k_{660}/k_{wind}$ values (Fig. 6a). While under conditions of strong positive and negative gradients, $k_{660}$ followed the wind-based relationship; smaller gradients (both positive and negative) showed the largest variability. Furthermore, at these relatively high wind-speed conditions ($U_{10N} > 8\,\mathrm{m\,s^{-1}}$), the water column was well mixed and, while some

scatter is observed, the largest values of $k_{660}$ seemed to be related with a deep mixed layer (Fig. 6c). The combined effect of $\Delta pCO_2$ and MLD, was strongly modulated by the seasonal patterns. During the winter, strong and persistent vertical mixing occurred along with positive $\Delta pCO_2$ (Fig. 3a). During the summer, shallower MLD and negative $\Delta pCO_2$ were observed. Consistently, the probability distribution function showed a higher probability of positive $\Delta pCO_2$ during the high wind speed regime (Fig. A1a) together with a wide distribution of MLD (Fig.A1b), suggesting winter conditions during a large proportion of the high wind-speed regime. On the contrary, higher probability of negative $\Delta pCO_2$ and shallower MLD depth occurred during low wind speeds.

In addition to $\Delta pCO_2$ and MLD, the characteristics of the wave field were—to some extent—associated to an enhanced gas exchange. In particular, the highest significant wave heights ($H_s$>1.5 m) were consistent with the largest values of $k_{660}/k_{wind}$ (Fig. 6e). Low values of wave steepness ($H_s/L_p$) were observed even at the highest wind speeds with maximum values of 0.06 (Fig. A2e), much lower than the theoretical wave breaking threshold (Stokes, 1880). While the small values of the wave age, $C_p/U_{10N}$, (Fig. A2f) indicated growth of the wave field caused by the wind forcing over the surface. Thus, suggesting the prevalence of locally generated waves (i.e. wind sea) at these wind speed conditions.

Atmospheric conditions such as atmospheric stability, relative humidity, and the total enthalpy flux were also associated to the gas exchange efficiency. In terms of the atmospheric stability, a clear enhancement on $k_{660}$ was observed during unstable conditions in comparison to $k_{wind}$ (Fig. 6b). Meanwhile, during neutral and stable conditions, no significant difference was observed between $k_{660}$ and $k_{wind}$. Relative humidity below 70 % were consistent with $k_{660}/k_{wind}$ >1, while during higher RH, the scatter in the data was low and $k_{660}$ was well represented by $k_{wind}$. Furthermore, an increasing trend was observed in terms of the total enthalpy flux, with higher $k_{660}$ values observed when high enthalpy fluxes occurred.

Based on the analysis presented in Fig. 6, we identified a set of conditions that were associated with enhanced values of $k_{660}$. These conditions were characterized by positive $\Delta pCO_2$, strong water-side mixing and dry air (RH < 70 %) during unstable atmospheric stratification. A wave field with $H_s$ >1.5 m, further enhanced the gas exchange. Gas transfer velocities higher than predicted, not only by $k_{wind}$, but also by other commonly-used parametrizations were observed under these specific conditions (Fig. 7). These enhanced conditions, were observed particularly during high wind speeds, but also during the intermediate regime, and to a much lesser extent during the low wind speed conditions. When these data were excluded from the analysis, $k_{660}$ was better represented by $U_{10N}$ following a quadratic relationship ($R^2$ =0.62). Furthermore, the enhanced $k_{660}$ (blue dots in Fig. 7), showed a wind-speed dependency of higher order (cubic) and $R^2$ =0.57.

### 3.2.2 Controls on $k_{660}$ under low wind-speed conditions

Under low wind-speed conditions ($U_{10N} \leq 6\,\mathrm{m\,s^{-1}}$), the scatter of the data was even larger than under higher wind speeds. Water-side parameters such as $\Delta pCO_2$, MLD, and water-side convection helped explain part of the variability observed. Gas transfer velocities higher than predicted by $k_{wind}$ were observed for both positive and negative gradients, showing that only under very strong negative gradients ($\Delta pCO_2$ < -162 $\mu$atm) the gas exchange was lower than expected by the wind-speed relationship (Fig. 8a). Under these calmer wind speed conditions, the values of MLD were generally low (Fig. A1b), with the lowest (MLD<15) values showing a larger scatter in $k_{660}$ (Fig.A1k). The pattern of $\Delta pCO_2$ and MLD was consistent with

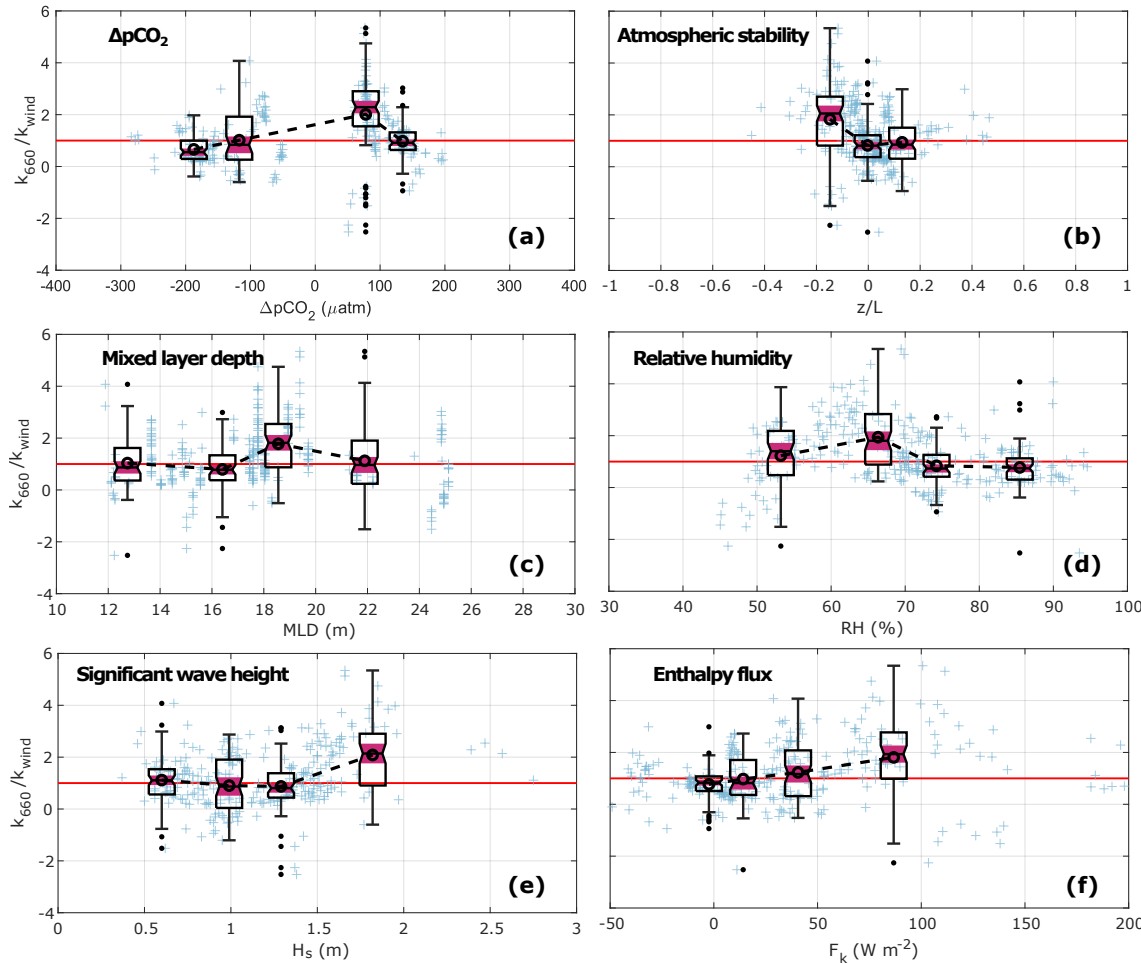

**Figure 6.** Normalized gas transfer velocity ($k_{660}/k_{wind}$) under high wind speed conditions ($U_{10N} > 8\ \mathrm{m\ s^{-1}}$) as a function of a) $\Delta pCO_2$, b) atmospheric stability ($z/L$), c) mixed layer depth (MLD), d) relative humidity (RH), e) significant wave height ($H_s$), and f) total enthalpy flux ($F_k$). The crosses represent the individual half-hourly values. The boxplots give a statistical summary for equidensity bins defined based on the distribution of $k_{660}/k_{wind}$ as a function of each of the parameters (see Appendix A). The median, first, and third quartiles are represented in each box; the whiskers represent the minimum and maximum values, and the black dots represent the outliers; the notches highlighted in pink indicate the median's 95 % confidence interval. The open circles linked with a dashed line indicate the bin means, and the horizontal red line indicates $k_{660}/k_{wind} = 1$.

the seasonal cycle where shallow MLD, typical of the summer months, can hinder the downward transport of $CO_2$ into the sea (negative $\Delta pCO_2$). The wave field showed smaller ($H_s$<1 m) and less steep waves. Under these conditions, the waves tended to be older ($C_p/U_{10N}$>1.2), indicating a larger proportion of swell waves in comparison to the locally generated waves observed at higher wind speeds. The characteristics of the wave field did not seem to induce any deviation of $k_{660}$ with respect
290   to $k_{wind}$ at low wind speeds (Fig. A2).

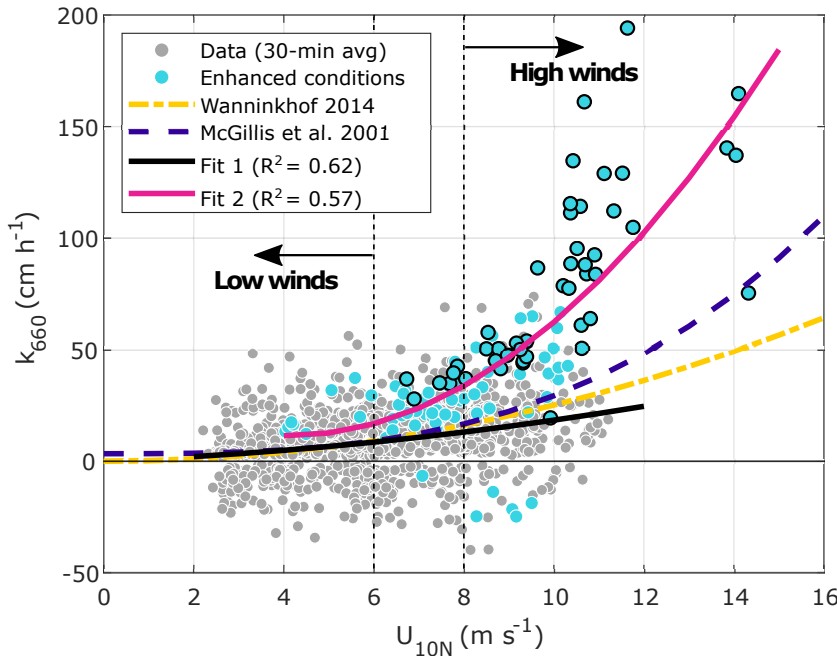

**Figure 7.** Gas transfer velocity for $CO_2$ (adjusted to a Schmidt number of 660) as a function of the $10\,\mathrm{m}$ neutral wind speed. The dots represent the half-hourly values of $k_{660}$. The blue dots represent $k_{660}$ under enhanced conditions (see text for details), while the blue dots with a black edge indicate cases where $H_s > 1.5\,\mathrm{m}$. The black line represents the best fit (quadratic) to the data excluding the enhanced cases (only gray dots), while the pink line is the best fit to the enhanced data (only blue dots). For reference, a quadratic (Wanninkhof, 2014) and cubic (McGillis et al., 2001) wind-based parametrizations were included. The wind speed regimes are separated by vertical dashed lines.

In addition to $\Delta pCO_2$ and the MLD, atmospheric stability and the water-side convective scale ($w*$, as defined in Jeffery et al. (2007)) were relevant parameters used to explain at least part of the variability in $k_{660}$. On average, lower $k_{660}$ than expected by $k_{wind}$ during unstable atmospheric conditions and high water-side convective scale were observed (Fig. 8b and c). However, large variability was observed under these conditions. Meanwhile, $k_{wind}$ seemed to better represent the behaviour of $k_{660}$ under neutral and stable conditions, as well as during lower magnitudes of the water-side convective scale. Under low wind-speed conditions, large enthalpy fluxes were associated with $k_{660}/k_{wind} < 1$ (Fig. 8d).

Further analysis of the data during low wind speeds and unstable atmospheric conditions showed that water-side convection can enhance the gas exchange. Particularly during winter, when persistent cooling of the sea surface was expected ($\Delta T > 0$), enhanced values of $k_{660}$ were observed at low and intermediate wind speeds associated to high values of $w*$ (Fig. 9a and 10). On the contrary, low values of $w*$ were predominant during the summer months, and linked to low values of $k_{660}$ (Fig. 9b and 10). Furthermore, some relatively high values of $w*$ were observed during the summer, associated to a large proportion of the negative $k_{660}$ values observed. Under neutral and stable atmospheric conditions, water-side convection is not a relevant mechanism (Rutgersson and Smedman, 2010; Rutgersson et al., 2011; Norman et al., 2013b).

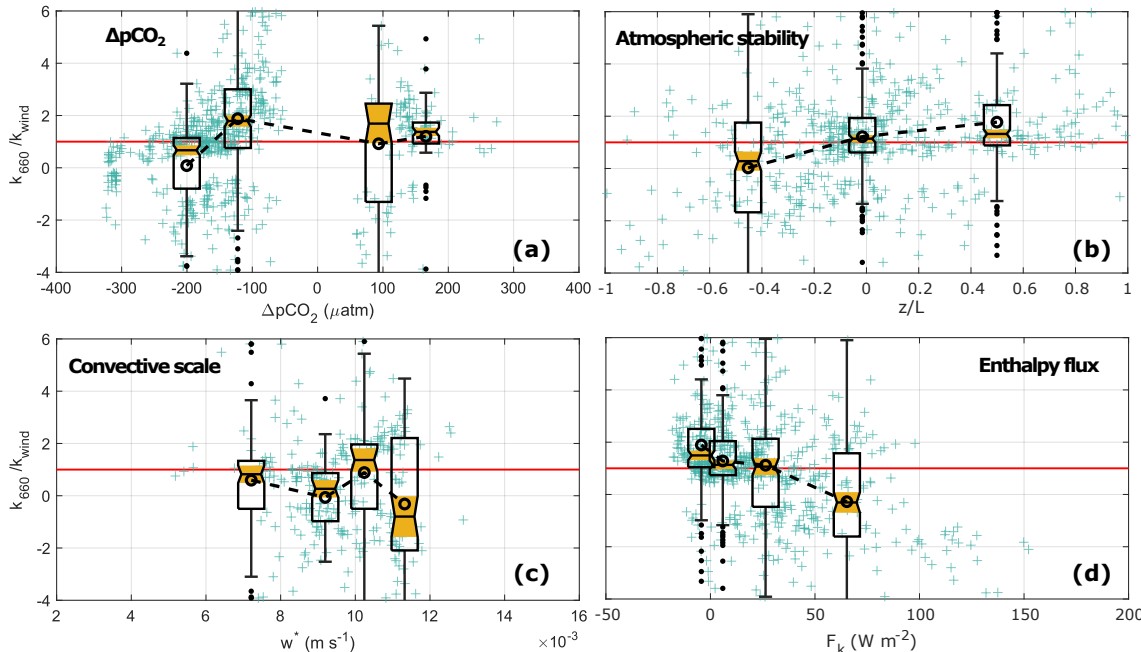

**Figure 8.** Normalized gas transfer velocity ($k_{660}/k_{wind}$) under low wind speed conditions ($U_{10N} \leq 6\,\mathrm{m\,s^{-1}}$) as a function of a) $\Delta pCO_2$, b) atmospheric stability ($z/L$), c) water-side convective scale ($w*$) under unstable atmospheric conditions, and d) enthalpy flux ($F_k$). The crosses represent the individual half-hourly values. The boxplots give a statistical summary for equidensity bins defined based on the distribution of $k_{660}/k_{wind}$ as a function of each of the parameters (see Appendix A). The median, first, and third quartiles are represented in each box; the whiskers represent the minimum and maximum values, and the black dots represent the outliers; the notches highlighted in yellow indicate the median's 95 % confidence interval. The open circles linked with a dashed line indicate the bin means, and the horizontal red line indicates $k_{660}/k_{wind} = 1$.

The rest of the scattered data which occurred during negative $\Delta pCO_2$ and shallow MLD cannot be explained by water-side convection or any other parameter assessed in this study. Furthermore, the interpretation of these data should be taken with some caution as the strong stratification, relatively weak $\Delta pCO_2$, and the possibility of strong heterogeneity in terms of the biogeochemical properties might hinder our capacity to calculate $k_{660}$ from $pCO_2^w$ and $FCO_2$.

### 3.2.3 Controls on $k_{660}$ in the intermediate wind-speed range

Wind speeds between 6 and $8\,\mathrm{m\,s^{-1}}$ were considered here as the intermediate range. As such, the characteristics of this intermediate wind-speed range were found to be a transition in terms of the physical conditions between the low and high wind speeds (see Figures in Appendix A) . The wave field showed waves with an average $H_s$ of $0.9\,\mathrm{m}$, in comparison to the 1.4 and $0.6\,\mathrm{m}$ of the high and low wind speed regimes, respectively. However, wave steepness and wave age at intermediate winds showed average values of $H_s/L_p$=0.03 and $C_p/U_{10N}$=1.1 similar to the mean values of $H_s/L_p$=0.03 and $C_p/U_{10N}$=1.0 observed at higher wind speeds. However, no clear effect of the wave field was observed on $k_{660}$ (Fig. A2g, h and i). Both the

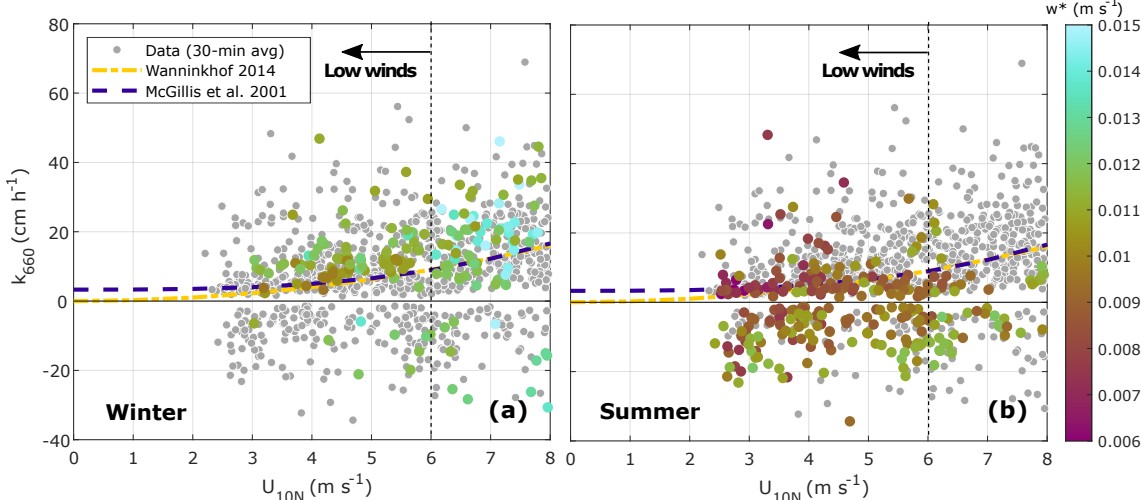

**Figure 9.** Gas transfer velocity for $CO_2$ (adjusted to a Schmidt number of 660) as a function of the 10 m neutral wind speed during a) winter and b) summer. The dots represent the half-hourly values of $k_{660}$. The color represents the water-side convective scale ($w*$) for data under unstable atmospheric conditions, calculated according to Rutgersson and Smedman (2010). The wind speed regimes are separated by a vertical dashed line.

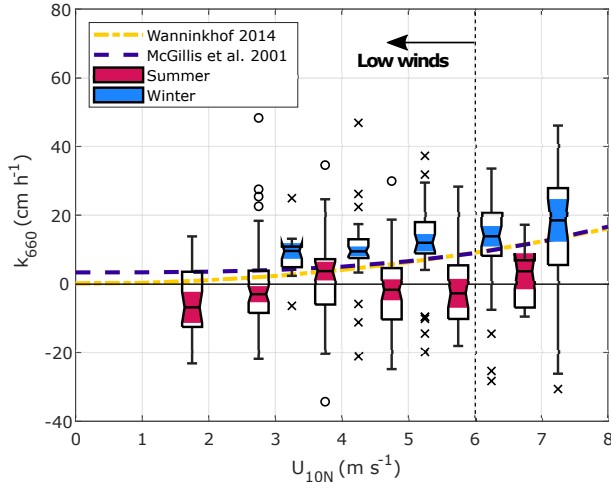

**Figure 10.** Boxplots of the gas transfer velocity for $CO_2$ (adjusted to a Schmidt number of 660) during unstable atmospheric conditions as a function of the 10 m neutral wind speed during summer (pink) and winter (blue). The median, first, and third quartiles are represented in each box; the whiskers represent the minimum and maximum values, and the circles and crosses represent the outliers; the notches highlighted in color indicate the median's 95 % confidence interval. The wind speed regimes are separated by a vertical dashed line.

wave field and the wind speed seemed to cause stronger mixing as larger values of MLD reaching the entire water column (ca 25 m) were observed under the transition range in contrast to the more persistent stratification at lower wind speeds (Fig. A1b).

Values of MLD during these intermediate wind speeds were still lower, on average, than those observed during the high wind speed regime. Meanwhile, large values of $w^*$ suggest the mixing caused by water-side convective processes during unstable atmospheric conditions might be relevant at the intermediate wind speeds, in particular, during the winter (Fig. 9a).

## 4 Discussion

We used nine years of eddy-covariance-based $FCO_2$ data to evaluate the effect of different control mechanisms on air–sea $CO_2$ gas exchange. By using this long record, we were able to capture the seasonal and inter-annual variability of the $FCO_2$ and other parameters relevant to the gas exchange (Sect. 3.1), as well as directly assess controls on $k_{660}$ (Sect. 3.2). Direct $FCO_2$ measurements over long periods are necessary in order to resolve the effect of multiple parameters on the gas exchange at both short- and long-term (several years) scales.

The empirically derived $k_{660}$ values showed, on average, a similar wind-speed dependency as two wind-based parametrizations, the quadratic relationship from Wanninkhof (2014) and the cubic relationship from McGillis et al. (2001) (Fig. 5). These parametrizations were used as references, and further comparison with the data presented here is beyond the scope of the study. Furthermore, we considered that the use of other commonly-used parametrizations (e.g. Wanninkhof and McGillis, 1999; Nightingale et al., 2000; Ho et al., 2006) would show similar results. Based on the results presented here, we showed, on the one hand, that a wind-based model with a similar pattern to those developed for larger basins and open-ocean conditions, was an accurate representation ($R^2 = 0.97$) of the mean $k_{660}$. On the other hand, the scatter of the half-hourly data suggested that a large proportion of the short-term variability in $k_{660}$ was not explained by variations in the wind speed. Previous studies have already pointed out the relevance of mechanisms other than the wind as controls on air–sea gas exchange in heterogeneous environments such as marginal and coastal seas (e.g. Upstill-Goddard, 2006; Gutiérrez-Loza et al., 2021). Here, based on the analysis of atmospheric and water-side parameters, we identified conditions associated with enhanced $k_{660}$ during high and low wind speed conditions.

We identified a set of conditions that were associated with enhanced values of $k_{660}$ at intermediate and high wind speeds (Fig. 7). These conditions were characterized by positive $\Delta pCO_2$ (i.e. winter conditions), unstable atmospheric stratification, strong vertical mixing in the water column (deep MLD), and low relative humidity. These conditions, affecting the upward fluxes during the winter months, resulted in an even more pronounced enhancement of $k_{660}$ when the wave field presented high significant wave heights ($H_s > 1.5\,\mathrm{m}$). We suggest here, that the high $k_{660}$ values observed can be potentially explained by the the effect of sea spray on air–sea $CO_2$ flux under the aforementioned conditions. In this hypothesis, the role of atmospheric parameters as control mechanisms is a key factor for the the otherwise water-side controlled $CO_2$ exchange across the sea surface. Unstable atmospheric conditions, for example, have been previously found to be associated with enhanced $CO_2$ transport (Andersson et al., 2019; Van Dam et al., 2021), possibly due to additional small-scale turbulence, as suggested by Andersson et al. (2019). However, in this study we found that none of the individual parameters had a clear effect on $k_{660}$, and was rather the combination of these parameters leading to the specific conditions enhancing the exchange, potentially due to effective evaporation of droplets with relatively high $pCO_2^w$.

The characteristics of the wave field are one of the most important parameters involved on the air–sea gas transfer directly affecting the conditions of the interface. However, the effect of waves on gas exchange has been very seldom studied in coastal regions (e.g. Gutiérrez-Loza et al., 2018). We analysed different parameters describing the wave field (significant wave height, wave steepness, and wave age) and their effect on the gas transfer velocity. At high wind speeds, the waves were found to be locally generated by the wind (i.e. wind sea), which made it hard to separate the effect of the waves from that of the wind. The wave climate in the Baltic Sea is characterized by predominant short wind sea and scarce long swell (Soomere, 2022). Furthermore, the effect of swell waves was not relevant at high wind speed conditions, and while at lower wind speeds some effect from swell could be expected, such effect was partly excluded here as wind speeds lower than $2\,\mathrm{m\,s^{-1}}$ were removed from the analysis. Values of wave steepness suggested that there was no systematic wave breaking. However, empirical knowledge of the study site suggests the presence of breaking, particularly of the smallest/youngest waves. These waves are not expected to generate significant bubble entrainment into the upper layer of the sea, but generation of sea spray is expected at the highest wind speeds.

The role of sea spray as a control mechanisms of the air–sea gas exchange has been reported before (Andreas et al., 2016, 2017). The impact of a broken surface caused by wave breaking and whitecapping, has been shown to cause an asymmetric transport due to the injection of bubbles into the ocean enhancing the negative (downward) fluxes (Woolf, 1997; Leighton et al., 2018). However, the potential effect of such processes on positive fluxes is still unclear. Here, we found that the increase in the $k_{660}$ was associated to the ocean-to-atmosphere (upward) transport in what we suggest is the effect of sea spray. This effect was clearly affected with the seasonal variability as the increased $k_{660}$ where observed during the winter months, when dry and cold air (colder than the ocean) are common (Fig.4c and e). During the winter months positive $CO_2$ gradients prevail and positive gas and heat fluxes are most often observed. High enthalpy fluxes were observed during the conditions enhancing $k_{660}$, however, the mechanisms through which the transport is enhanced might be different as heat and moisture are air-side controlled. Nonetheless, the increased evaporation rates most probably found during low relative humidity and unstable conditions might also play a role on $CO_2$ transport. Dedicated analysis of the effect of bubbles and sea spray in coastal regions are still necessary. Particularly at high latitudes, where strong seasonality may cause significant differences in the forcing mechanisms modulating the influx and eflux of gas into/from the ocean (see Sect. 3.1). Further data under these particular conditions is necessary to establish a better understanding of the effect of sea spray mediated gas fluxes, as it can be a relevant mechanism modulating $FCO_2$ at global scales.

Under calm wind speed conditions, with $U_{10N} < 6\,\mathrm{m\,s^{-1}}$, the effect of water-side convection was found to be relevant to explain the larger values of $k_{660}$ during unstable atmospheric conditions at these wind speeds. Previous studies have shown the effect of convective processes on the gas transfer in the Baltic Sea (Rutgersson and Smedman, 2010; Norman et al., 2013b). The same studies showed that the water-side convective processes are not relevant under stable and neutral conditions, as well as at wind speeds higher than $8\,\mathrm{m\,s^{-1}}$. Here, we found the effect of water-side convection to be strongly seasonal (Fig. 9 and 10), were a clear enhancement due to water-side convection occurred during the winter. During the summer, low values of $w^*$ seemed to mostly follow the wind-speed relationships. However, intermediate values of $w^*$ can be linked with most of the negative $k_{660}$ observed during low and intermediate wind speeds (Fig. 9b). Thus, indicating that it was during these

385 unstable conditions when the mismatch between the observed fluxes and $\Delta pCO_2$ occurred. Further investigation is necessary in order to evaluate the effect of water-side convection at other temporal scales (i.e. diurnal variability), as well as the potential implications on $FCO_2$ at larger spatial scales.

In addition to the processes analysed in this study, there are other mechanisms which are relevant for the gas exchange in the Baltic Sea at different temporal and spatial scales. The occurrence of upwelling events can be up to 25-30% in some regions of 390 the Baltic Sea (Lehmann et al., 2012). Norman et al. (2013a) found that the effect of these events on the annual mean $FCO_2$ can reach up to 25% for the Baltic Sea conditions. Upwelling is not expected to be relevant for the wind directions analyzed in this study, however, we recognize that the effect of upwelling in the area can be important and can have implication in the observed inter-annual variability. Furthermore, using the parametrization suggested by Pereira et al. (2018) based on the skin temperature, we calculated a rough estimate of the effect of surfactants on our long-term $k_{660}$. The results suggest an 395 overall effect of merely -0.1%, suggesting a very small reduction of $k_{660}$ when surfactants are taken into account. A detail analysis of the effect of surface films is beyond the scope of this study. Nevertheless, we recognize that this process might be particularly relevant in coastal seas and other shallow bodies of water. The analysis of the effect of sea ice, precipitation, Langmuir circulation, and other processes in the Baltic Sea is still pending.

Eddy covariance data has been increasingly used in marine environments to study air–sea gas exchange (e.g. McGillis et al., 400 2001; Miller et al., 2010; Andersson et al., 2016; Gutiérrez-Loza et al., 2019). Although eddy covariance measurements have greatly contributed to the better understanding of gas exchange processes, some caveats still exist regarding this methodology. Dong et al. (2021) recently published a thorough analysis regarding the uncertainties associated to air–sea eddy covariance and subsequent gas transfer velocity calculations. In that study, the total uncertainty was calculated as $\delta F = (\delta F_S^2 + \delta F_R^2)^{0.5}$, where $\delta F_R$ is the random error component and $\delta F_S$ is the systematic bias. According to their estimates, the total propagated 405 bias (estimated from the individual sources) represented a flux uncertainty of 7.5%. The largest potential source of bias was the motion of the ship, followed by the imperfect calibration of the sensors and the effect of the inlet tubes (i.e. for close-path gas analyzers); other potential sources of bias such as airflow distortion, instrument separation, and water vapour cross-sensitivity were found negligible. An additional 5% bias due to insufficient sampling time resulted in a total systematic error of 9%. Furthermore, Dong et al. (2021) found that the contributions of the random errors ($\delta F_R$) represented a much larger 410 contribution (20–50%) to the total flux uncertainty, with the low $CO_2$ flux regions presenting the largest uncertainty. According to Dong et al. (2021), the total flux uncertainty ($\delta F$) represents 10% of the variance in $k_{660}$. Moreover, using data from the Östergarnsholm station, Rutgersson et al. (2008) estimated an uncertainty of 17% associated to the eddy covariance fluxes (using two open-path gas analyzers), while the contributions of the other terms involved in the gas transfer velocity calculation, i.e. solubility parameter and $\Delta pCO_2$, were estimated to be 1% and 4%, respectively. The total instrumental errors estimated 415 in Rutgersson et al. (2008), which do not consider any methodological biases, resulted in a total uncertainty of 20% in the gas transfer velocity estimates. Assuming that the values presented by Dong et al. (2021) are applicable to our study, and considering that the land-based station is not subject to platform movement or the effect of inlet tubes, a rough estimate of the flux uncertainty caused by systematic bias gives a $\delta F_S < 6.4\%$. Furthermore, based on the monthly mean $FCO_2$ (Fig. 3), Östergarnsholm is on the "high $CO_2$ flux" regime ($> 10\,\mathrm{mmol\,m^{-2}\,d^{-1}}$) during most of the year, therefore laying on the lower

range ($\sim 20\,\%$) of the random uncertainty values. Based on these values of random error and systematic bias, a total uncertainty value of $\sim 20\,\%$ associated to the flux measurements can be assumed. The propagated error associated to the gas transfer velocity is, therefore, also in the order of $\sim 20\,\%$, in agreement with previous estimates from Östergarnsholm (Rutgersson et al., 2008).

In addition to the uncertainties discussed above, in this study we identified two major limitations that further affected the analysis. First, the large amount of data that was removed from the analysis due to quality control (Sect. 2.2 and Appendix B). The criterion excluding the largest amount of data was the wind-direction selection (open-sea sector), criterion that excluded ca 85\,% of the total data (Table B1). Relaxing this criterion would significantly increase the amount of data used for the $k_{660}$ analysis. However, it would also mean including fluxes associated with more heterogeneous regions where the measured water properties might not necessarily be representative of $FCO_2$ (see Rutgersson et al., 2020); hence, larger uncertainties in the $k_{660}$ estimates. Furthermore, the criterion regarding the signal quality of the gas analyser ($\sigma^2_{RSSI} <0.001$) was only fulfilled by 36.7\,% of the flux data. Most of the data that was removed due to the low signal criterion occurred during precipitation events or very high wind speeds, when droplets land on the optical windows of the gas analyser. Gutiérrez-Loza et al. (2021) found that precipitation (rain) increased the net $CO_2$ uptake by 4\,% in the Baltic Sea during 2009-2011, while Ashton et al. (2016) found that the effect of rain can lead to a 6\,% increase in the ocean uptake, globally. By removing the data due to low signal strength, the effects of precipitation remained unaccounted for in the analysis presented here, as well as fluxes under high wind speed conditions (above 12–14 $\mathrm{m\,s^{-1}}$), were strong mixing, bubble injection, and sea spray might be important controls on air–sea gas exchange (e.g. Blomquist et al., 2017; Bell et al., 2017). Removing small fluxes (i.e. $|FCO_2| < 0.05\,\mathrm{\mu mol\,m^{-2}\,s^{-1}}$) due to the gas analyser detection limit, might bias the net flux estimates. Therefore, we limited this study to the analysis of the half-hourly values and general seasonal patterns; however, conclusions about whether the region was an overall sink or source of $CO_2$ were avoided. Finally, the potential effect of swell on the $CO_2$ exchange at very low wind speeds was not accounted for, given that wind speeds lower than 2 $\mathrm{m\,s^{-1}}$ were removed from the analysis. The other criteria (see Table B1) were expected to cause very small biases, if any, on the data. The probability distribution of the quality controlled data set showed similar patterns compared to the initial data set (before quality control) for each parameter used in the analysis (i.e. $\Delta pCO_2$, wind speed, wave properties, etc.). Hence, indicating that the reduced data set used for the analysis was a good representation of the conditions observed in the study area over the nine-year measurement period.

The second caveat was the large amount of negative $k_{660}$ observed (Fig. 5). These data fulfilled every step of the quality control process, and therefore, there was no methodological reason to exclude these values from the data set. However, further investigation is needed to understand the source of these data and whether or not there is a viable physical explanation. We considered that part of the negative values observed here can be attributed to the inherent turbulent characteristics of eddy covariance measurements. Furthermore, at low wind speeds, conditions hindering $k_{660}$ calculations (e.g. strong stratification), as well as instrumental limitations and set-up might, such as $pCO_2^w$ measurements response time and depth, might become relevant. Furthermore, processes not accounted for such as chemical enhancement and surfractants (e.g. Ribas-Ribas et al., 2018), might be also playing a role on the amount and distribution of negative $k_{660}$ observed.

Regardless of the length of our data set, we still found three major limitations when addressing the effect of forcing mechanisms on air–sea gas exchange: 1) the use of a land-based station provides significant advantages, but it hinders the evaluation of the spatial variability of the exchange; 2) the intrinsic characteristics of the eddy covariance technique, in combination to the exclusion of wind directions not representative of open-sea conditions, resulted in a patchy data set with significant gaps throughout the time series; and 3) limited amount of data at high wind speeds ($U_{10N} > 10\,\mathrm{m\,s^{-1}}$) was recorded, which hindered the analysis and increases the uncertainties at those critical wind conditions. The results presented here provide significant insights on the air–sea gas exchange and its variability. Nevertheless, the limitations are still large and a continued effort from the scientific community is required.

Global coastal oceans and marginal seas are one of the most vulnerable environments subjected to the effect of climate change and anthropogenic pressures (Pachauri et al., 2014). Understanding the role of these regions in the global carbon cycle has become an essential aspect in order to address the challenges of the current and future climates. In this sense, the Baltic Sea can be seen as a test basin which provides a wide variety of physical and biogeochemical conditions. At the same time, the carbon system of the Baltic Sea has been relatively well-documented (e.g. Kuliński and Pempkowiak, 2011; Schneider and Müller, 2018), and the region has been a relevant study area in terms of mitigation and environmental management (Reusch et al., 2018). The analysis and results presented here can be relevant for other marginal seas and coastal areas. While the potential effect of sea spray and water-side convection is certainly relevant at global scales, and further investigation is encouraged.

## 5   Conclusions

We presented a large data set of directly measured air–sea $CO_2$ fluxes by eddy covariance from a land-based station in the Baltic Sea. The forcing mechanisms acting on the surface of the ocean and their relative effect on the gas exchange can widely vary depending on the wind-speed. Therefore, the air–sea gas exchange, controlled by such forcing mechanisms, can also be expected to be affected in different ways depending on the wind-speed regime. We investigated the effect of the water-side and atmospheric conditions on the gas transfer velocity under relatively high, intermediate, and low wind speed regimes.

At high wind speeds ($U_{10N} > 8\,\mathrm{m\,s^{-1}}$), large $k_{660}$ values were observed in what we identified to be cold and dry air, under unstable atmospheric conditions. We suggest, based on these results, that sea spray might be one of the most effective mechanisms enhancing air–sea $CO_2$ fluxes under these conditions, partly due to large evaporation rates. The effect of the wave field was particularly evident in terms of the significant wave height, with high gas transfer velocity values occurring when large values of $H_s$ were observed. However, the effect of the wave field was not completely decoupled from the effect of the wind given that most of the waves were locally generated. Under low wind speed conditions ($U_{10N} < 6\,\mathrm{m\,s^{-1}}$), water-side convection was the only parameter explaining part of the variability on $k_r$, particularly during the winter. Intermediate wind speeds showed a mixed behaviour, thus, we defined these wind speeds as a transition range. Under these intermediate wind speed conditions, the effect of sea spray is still relevant, similar to the behaviour at higher winds. While convective processes enhanced $k_{660}$ during winter, as it does at lower wind speeds.

A wind-based model, showing a similar pattern as currently existing wind-based parametrizations, showed to adequately represent the average behaviour of $k_{660}$. However, further investigation of parameters affecting the seasonal and inter-annual variability of the fluxes is needed to improve our understanding of air–sea gas exchange and adequately represent $k_{660}$ at shorter time scales. Similarly, a detailed analysis of bubble- and sea-spray-mediated fluxes is needed to contribute to the understanding of $CO_2$ fluxes in the coastal regions, and other heterogeneous environments where the asymmetric behaviour of the transport might have strong implications.

*Data availability.* The raw data supporting the conclusions of this manuscript is available upon request.

## Appendix A: Normalized gas transfer velocity

The full set of figures showing the normalized gas transfer velocity ($k_{660}/k_{wind}$) vs each of the parameters analysed in this study, are presented here. The parameters are divided in water-side control mechanisms (Fig. A1), wave field characteristics (Fig. A2), and atmospheric control mechanisms (Fig. A3). Each figure also include the probability distribution function (PDF) of each variable for high, intermediate, and low wind speeds to highlight the difference between the wind regimes.

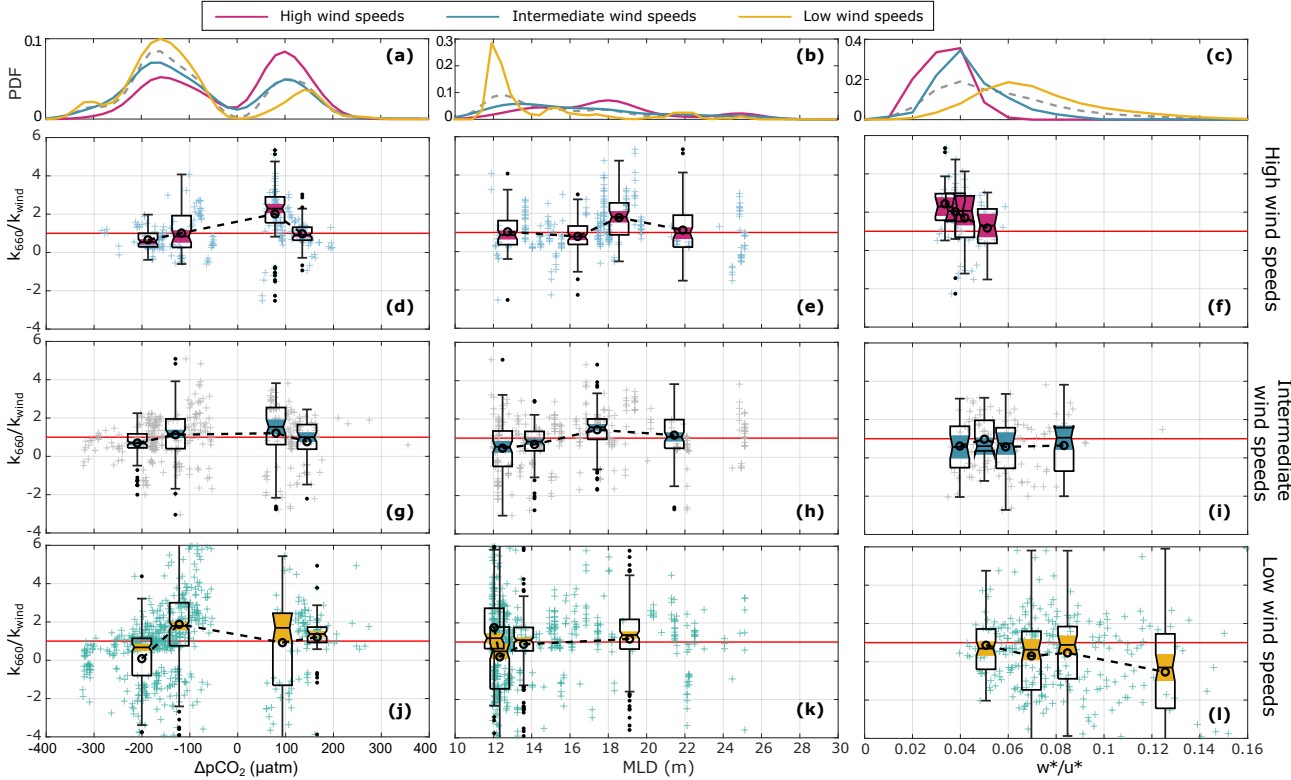

**Figure A1.** Water-side control mechanisms: $\Delta pCO_2$ (left), mixed layer depth (center), and water-side convective scale normalized with the friction velocity ($w*/u*$) (right). Top panels a),b), and c) show the probability distribution function (PDF). Panels d) to l) show the normalized gas transfer velocity ($k_{660}/k_{wind}$) under high (upper), intermediate (middle) and low wind speeds (lower). The crosses represent the individual half-hourly values. The boxplots give a statistical summary for equidensity bins defined based on the distribution of $k_{660}/k_{wind}$ as a function of each of the parameters. The median, first, and third quartiles are represented in each box; the whiskers represent the minimum and maximum values, and the black dots represent the outliers; the notches highlighted in yellow indicate the median's 95 % confidence interval. The open circles linked with a dashed line indicate the bin means, and the horizontal red line indicates $k_{660}/k_{wind} = 1$.

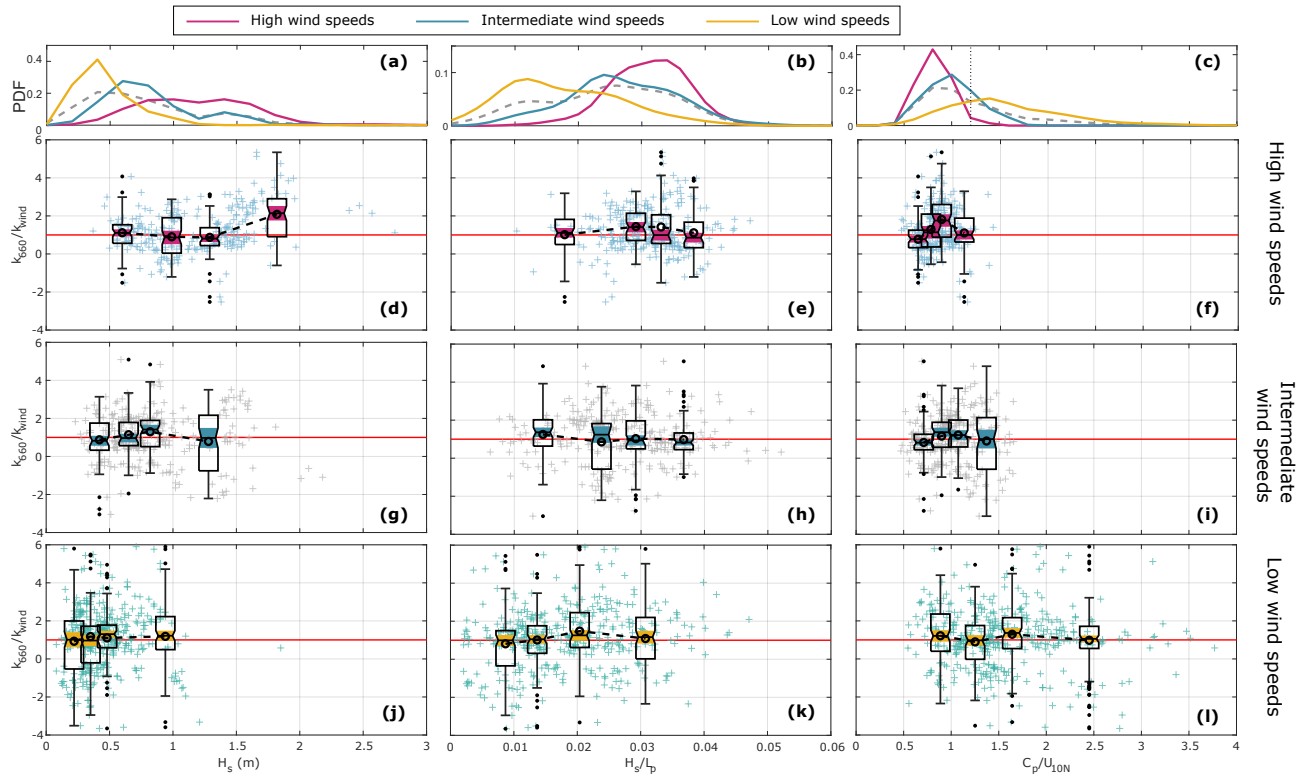

**Figure A2.** Same as in Fig. A1 for wave-field characteristics: significant wave height (left), wave steepness (center), and wave age (right).

## Appendix B: Quality control statistics

During the nine-year record between 2013 and 2021, we recorded 125,001 $FCO_2$ data points and 66,475 $k_{660}$ data points out of the maximum possible of 157,776 (half hours in nine years). However, strict quality control and post-processing analysis was necessary to ensure that only high-quality data was used for the flux calculations using the eddy covariance method (See Sect. 2.2). A significant amount of data was removed from the data set based on these criteria. Information about the relative importance of each criterion of the quality control is presented in Table B1.

The wind direction selection criterion (i.e. using only the open-sea sector) is the procedure that rejects the largest amount of data from the data set (Table B1). Considering that the total amount of data for this sector, before the quality control, was 18,625 $FCO_2$ and 8,974 $k_{660}$ data points. Thus, the final data set consisted of 18.7 % and 15.0 % of the open-sea initial data for $FCO_2$ and $k_{660}$, respectively; This corresponded to 3,477 and 1,349 data points, respectively.

The wind direction selection (i.e. using only the open-sea sector) is the procedure that rejects the largest amount of data from the data set (Table B1). Considering that the total amount of data for this sector was 18,625 $FCO_2$ and 8,974 $k_{660}$ data points before quality control, the final data set consisted of 18.7 % and 15.0 % of the open-sea initial data for $FCO_2$ and $k_{660}$, respectively. This corresponded to 3,477 $FCO_2$ and 1,349 $k_{660}$ data points.

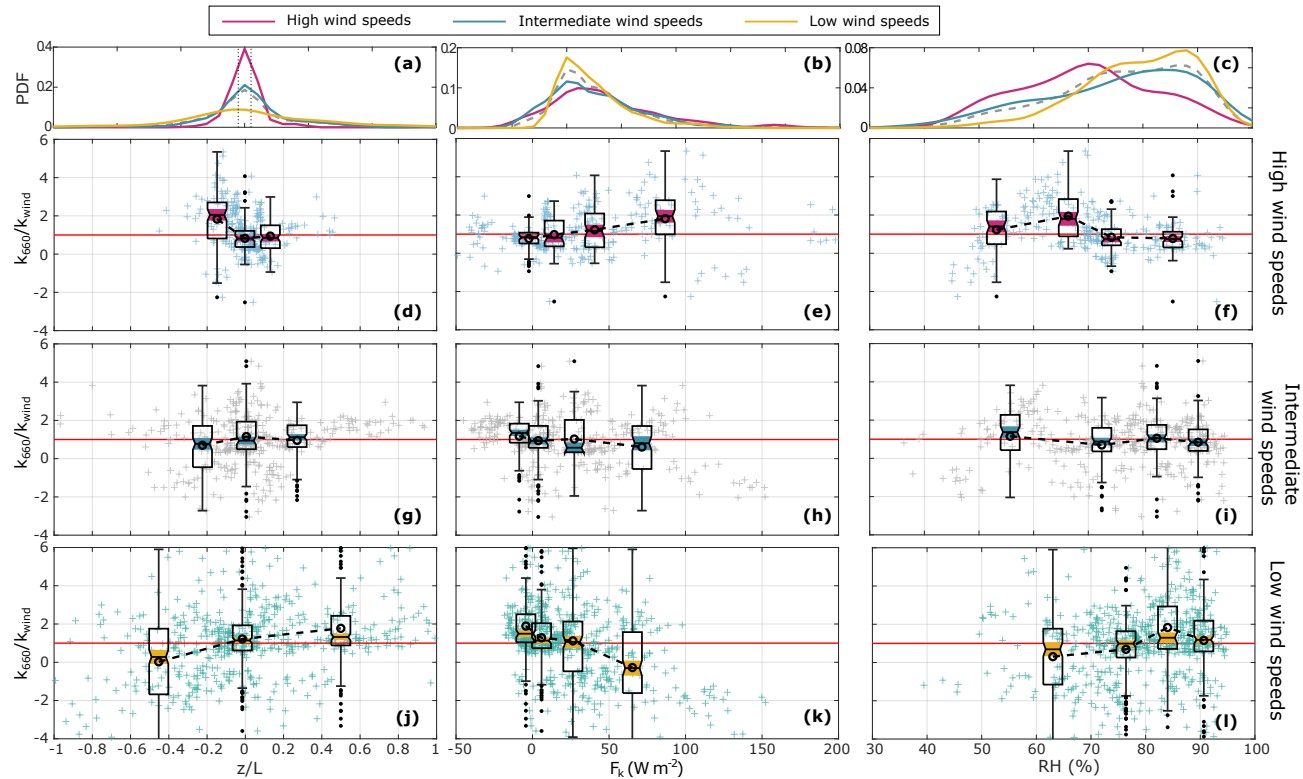

**Figure A3.** Same as in Fig. A1 for atmospheric control mechanisms: atmospheric stability (left), total enthalpy flux (center), and relative humidity (right)

*Author contributions.* LG-L, MW, ES and AR were involved in the conceptualization of the project. AR was in charge of the project administration and funding acquisition. The formal analysis was carried out by LG-L with the support of EN. LG-L, MW, EN and AR participated in the continuous maintenance of the Östergarnsholm station and data acquisition. LG-L wrote the original draft with contribution from all the co-authors.

*Competing interests.* The authors declare that they have no conflict of interest.

*Acknowledgements.* The ICOS station Östergarnsholm is funded by the Swedish Research Council (Vetenskapsrådet) grants 2012-03902 and 2013-02044, and Uppsala University. The wave data was provided by Heidi Pettersson from the Finish Meteorological Insitute (FMI) to whom the authors are grateful.

**Table B1.** Percentage of data that successfully fulfill each individual quality control criterion. The percentages are relative to the total recorded amount of data ($100\% = 125{,}001$ for $FCO_2$ and $100\% = 66{,}475$ for $k_{660}$).

| Quality control criteria | $FCO_2$ data (%) | $k_{660}$ data (%) |
|---|---|---|
| $U_{min} = 2\,\mathrm{m\,s^{-1}}$ | 95.1 | 94.5 |
| Signal quality ($\sigma^2_{RSSI} < 0.001$) | 36.7 | 41.7 |
| Turbulence level ($\sigma^2_w > 1e^{-6}\,\mathrm{m^2\,s^{-2}}$) | 99.4 | 99.4 |
| Remove outliers | 80.0 | 81.7 |
| $|FCO_2|_{min} = 0.05\,\mathrm{\mu mol\,m^{-2}\,s^{-1}}$ | 51.4 | 48.4 |
| $|\Delta pCO_2|_{min} = 50\,\mathrm{\mu atm}$ | N/A | 41.3 |
| RH $< 95\,\%$ | 89.2 | 90.3 |
| Open-sea sector ($80° < \mathrm{WD} < 160°$) | 14.9 | 13.5 |
| $|\Delta T_w|_{max} = 1\,°\mathrm{C}$ | N/A | 77.6 |

N/A = Not applicable. The corresponding criterion has no impact on the resulting amount of data.

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
