# Peer review of "On physical mechanisms enhancing air-sea CO2 exchange"

_Biogeosciences, 2022_

## Referee Comment (RC1)

In this submission by Gutiérrez-Loza et al (GL), a lengthy dataset of eddy covariance data is presented, and used to probe the physical processes driving gas transfer in shallow coastal waters. In particular, results are separated into low and high wind-speed regimes, in which measured gas transfer velocity (k) is correlated with various environmental drivers. GL finds that water-side convection dominates k at lower wind speeds, but combined wind, wave, sea spray, and other forcing take when wind exceeds 8 m/s. My general impression of GL is that the methods are appropriate and largely well described (could use a few clarifications as discussed below), and the discussion and major conclusions are largely supported by the analysis conducted. I also find the scope of the work to be appropriate for publication in BG. Provided that the authors can address the primary concern raised below, and make some improvements to the methods (as also described below), I feel that this manuscript can make a nice contribution to the gas transfer literature.

I have two primary concerns about the work presented by GL:
- First, I am not certain that residual k (k_r) formulated in this manner (k_measured – k_W14) indeed has the effect of "removing the wind-speed dependency from k_660" (line 177, interpretations thereafter). This is because the W14 parameterization was developed for the ocean basin scale, using a (revised) estimate of the bomb 14C inventory and a global wind product. As described in the *comments and recommendations* section of W14, this formulation is intended to be used for "regional-to-global flux estimates of CO2". The W14 formulation is also intended for longer (multiple hours) time scales by squaring wind speed and averaging over 6+ hours, so I am not sure this is appropriate to compare with the 30-minute averaging intervals used in GL. Put simply, W14 describes the global relationship between wind and k, but does not necessarily *isolate* the effect of wind speed on k (except for the y-intercept which is forced through 0). Therefore, while k_r as formulated should correct for *some* of the wind-speed dependency from k_660, I do not feel that it can do so comprehensively at the time-scale applied here.
  With this being said, I am open to the use of k_r in this way, provided that the authors 1) disagree with my explanation above and can provide a reasonable rebuttal explaining so, or 2) if they decide to revise the manuscript text to verbally describe this issue, or 3) if they can calculate a new k_r in a way that better incorporates the uncertainties in the wind-based parameterization (W14 here).

- Secondly, it is not clear what benefit was gained by working with such a long (nearly a decade) EC dataset, as the correlation-based analysis of GL is similar to those applied to shorter-term datasets from the same measurement platform. I understand that a detailed time-series analysis was beyond the scope of this work, but a short discussion may be useful. So, maybe the authors can offer some advice as to the time required to capture the full range in gas transfer variability? i.e., for readers planning a similar coastal EC deployment, is it enough to measure for a year, or do we need many years to capture the variation in physical forcing described in GL?

Line-by-line comments:

92: I see that instruments were located 9m above the ground surface, but how high is this above the sea surface?

98: Was z/L uniform across wind directions within the southeast window?

110: Are there ancillary measurements of T (e.g. from a shaded thermometer, or maybe a closed-path IRGA) to show whether or not solar heating of the sonic anemometer affected the Ts record?

Data processing: I would like to see more detailed statistics showing how many 30-min datapoints were rejected according to individual screening criteria. Comparing, for example, figure 5 with the full time-series in figure 4, it appears that a large majority of data failed the screening criteria. If so, this needs to be fully explained in the methods.

174: The generation of excessive negative k values is a frequent criticism, and major caveat, of EC-based gas transfer studies. So, what is the justification for removing -k values when they otherwise meet the screening criteria applied to the rest of the dataset? Doesn't removing negative values artificially decrease the variability in calculated k?

177: As per the discussion above, I do not agree that $k_r$ "remov[es] the wind-speed dependency from $k_{660}$". Given that the majority of the analysis in GL revolves around $k_r$, I think some additional description of the W14 parameterization and it's applicability to the current study site is warranted.

245: This enhanced wind dependence of k under unstable conditions is consistent with prior work in the Baltic (https://doi.org/10.1007/s10546-018-0408-9) and elsewhere (https://doi.org/10.1007/s10546-018-0408-9; https://doi.org/10.1002/lno.11620). Since these conditions are associated with the largest deviation of measured $k_{660}$ from $k_{W14}$, can the authors offer any further ideas as to the major driving cause?

259: Couldn't the lack of relationship between the wave field and $k_r$ be in part explained by the fact that (as explained by the authors), the waves here are not swell but rather locally-generated by wind? I.e., one would expect a strong correlation between wind and wave height here?

299: Maybe I missed it, but I do not see where the formulation of McGillis 2001 is compared with the k values calculated by GL.

---

## Community Comment (CC1)

Comments Bernd Jähne, Heidelberg University to the preprint

**On physical mechanisms controlling air–sea CO2 exchange**

by

*Lucía Gutiérrez-Loza, Erik Nilsson, Marcus B. Wallin, Erik Sahlée, and Anna Rutgersson*

https://doi.org/10.5194/bg-2022-82

**1. Empirical correlation instead of analysis of mechanisms**

The authors claim in their paper to analyze the physical mechanisms controlling air-sea gas transfer beyond wind speed using a nine year record of continuous eddy covariance measurements taken at the Östergarnsholm tower. Such a long time series is, of course, a treasure, but the title of the paper is misleading in my view. The paper does just list possible other mechanisms than wind wind (or more precisely wind stress at the ocean surface) as the main driving force, but does not discuss at all any possible conceptual model to be checked by the data. All they do are empirical correlations with some other parameters. In addition, even an observed correlation does not mean that another driving force is found, because the investigated parameter might be correlated to another, which is the real driving force.

Therefore a more appropriate title would be "Empirical correlations of the gas transfer velocity with other parameters than wind speed" or a similar wording.

**2. The statistical analysis is not convincing**

- The authors use a multistep scheme to select only reliable measurements. But they do not specify which fraction is remaining. It is important to know to which extent specific conditions are excluded from the analysis. If the exclusion were large, this may severely bias any averaging and correlations.
- For the averages shown in Fig. 5 all, also negative gas transfer velocities have been used, but not in the following more detailed analysis. This is inconsistent. Either you rely data or not at all.
- The concept of the residual gas transfer velocity and the correlation with a single other parameter seems not to be a good choice. A theoretically more sound approach would be to use a multiparameter space approach with all possible parameters and then perform a principal component analysis or similar approach, as it is standard in pattern recognition and classification. Then it would be possible to identify the most important parameters and the degree of correlation between the parameters quantitatively and whether the results are statistically relevant. The approach of the authors is just qualitatively.

**3. Claimed effect of sea spray in enhancing air-sea CO2 fluxes questionable**

The authors claim that "sea spray can enhance the transport of CO2 across the interface, as it does with heat and moisture". This is an oversimplification, because the transport of heat and water vapor is air-side controlled, while the transport of CO2 is water-side controlled. Therefore a much more detailed analysis is required. Droplets may partly be a dead surface for gas transfer in the same way as those bubbles, which come in a much shorter time scale into equilibrium with the surrounding water than their life time. However, the influence of solubility is inverse. Droplets are efficient for highly soluble tracers.

---

## Author Comment (AC1)

MS No.: bg-2022-82 Authors: Lucía Gutiérrez-Loza et al. https://doi.org/10.5194/bg-2022-82

**Authors' reply to Referee #1 Bryce Van Dam**

**Dear Bryce,**

We would like to thank you for taking the time of reviewing this manuscript and for the very thorough and useful comments provided. We have made our best to address each of them, and we consider the manuscript to have improved significantly based on your feedback. Please find below the responses to each of your comments.

The response to each comment is written in blue italics, while the changes made in the revised manuscript are in red.

**Primary concerns:**

First, I am not certain that residual k (k\_r) formulated in this manner (k\_measured  $-k_W14$ ) indeed has the effect of "removing the wind-speed dependency from k\_660" (line 177, interpretations thereafter). This is because the W14 parameterization was developed for the ocean basin scale, using a (revised) estimate of the bomb 14C inventory and a global wind product. As described in the comments and recommendations section of W14, this formulation is intended to be used for "regional-to-global flux estimates of CO2". The W14 formulation is also intended for longer (multiple hours) time scales by squaring wind speed and averaging over 6+ hours, so I am not sure this is appropriate to compare with the 30-minute averaging intervals used in GL. Put simply, W14 describes the global relationship between wind and k, but does not necessarily isolate the effect of wind speed on k (except for the y-intercept which is forced through 0). Therefore, while k\_r as formulated should correct for some of the wind-speed dependency from k\_660, I do not feel that it can do so comprehensively at the time-scale applied here.

With this being said, I am open to the use of k\_r in this way, provided that the authors:

- 1) Disagree with my explanation above and can provide a reasonable rebuttal explaining so, or
- 2) If they decide to revise the manuscript text to verbally describe this issue, or
- 3) If they can calculate a new k\_r in a way that better incorporates the uncertainties in the wind-based parameterization (W14 here).

We understand the concerns regarding the suitability of W14 to remove the "wind-speed dependency" from our data and we agree with the explanation given. Therefore, we have reconsidered the use of a residual gas transfer velocity  $(k_r)$  for the analysis. In the revised manuscript, the W14 parametrization is only shown as a reference but not included as part of the

calculations. Instead, we used a normalized gas transfer velocity defined as  $k_{660/k_wind}$ , where  $k_wind$  was obtained from the cubic relationship fitted to the equidensity bin averages of the data set of the current study (shown in pink dashed line **Figure 5**). This normalized gas transfer velocity was used, then, to identify conditions with gas transfer velocities higher-than-expected solely by the  $k_{660}$  vs U10N relationship under the different wind speed regimes. Subsequent analysis was carried out on  $k_{660}$  itself.

**Updated version of Figure 5:**

Figure 5. Gas transfer velocity for CO2 (adjusted to a Schmidt number of 660) as a function of the 10 m neutral wind speed. The grey dots represent the half-hourly values of  $k_{660}$  for the nine-year period from 2013 to 2021. The black dots and bars, represent the  $k_{660}$  mean values and standard deviations, respectively, calculated for equi-density bins based on the wind speed percentiles; the best fit to the means is shown as the pink dotted line ( $k_{wind}$ ). For reference, a quadratic (Wanninkhof, 2014) and cubic (McGillis et al., 2001) wind-based parametrizations were included. The colors in the shaded area represent the data density (in counts) with a grid bin size of 1 m s-1 by 10 cm h-1.

**Data processing section (2.2). Information regarding k\_r was removed and the following paragraph was included:**

"The calculated k\_660 were used to study the effect of water-side and atmospheric control mechanisms on air-sea CO\_2 exchange. A wind-speed relationship (k\_wind) was calculated as the cubic (best) fit to the bin-averaged k\_660, using equidensity bins of the wind speed, and used to obtain a normalized gas transfer velocity defined as k\_660/k\_wind."

Corresponding changes in the **Results** section, from **sub-section 3.2** and onwards, as well as in the **Discussion**. Particularly, Figures 6 to 10 were removed and substituted by the following figures, and corresponding description and discussion (not presented in full here). **Figure 6** shows the normalized gas transfer velocity vs. several parameters for the high wind speed regime, where enhanced values of k\_660 can be identified (the full set of figures was included as

**Appendix** A in the revised manuscript). Based on this, we selected a data set with particular conditions: positive  $\Delta pCO2$ , strong mixing, high significant wave height, dry air and unstable atmospheric stratification. This feature set was found to be associated with higher k\_660 than predicted by the wind speed relationships (**Figure 7**) in what we suggest can be the effect of sea spray. A brief excerpt from the text presenting these results, reads as follows:

"Based on the analysis presented in Fig. 6, we identified a set of conditions that seemed to be associated with enhanced values of k\_660. These conditions were characterized by positive  $\Delta pCO_2$ , strong water-side mixing and dry air (RH < 70 %) during unstable atmospheric stratification. A wave field with H\_s >1.5 m further enhanced the gas exchange. Gas transfer velocities higher than predicted, not only by k\_wind, but also by other commonly-used parametrizations were observed under these specific conditions (Fig. 7). These enhanced conditions, were observed particularly during high wind speeds, but also during the intermediate regime, and to a much lesser extent during the low wind speed conditions. When these data was removed from the analysis, k\_660 seemed to be better represented by U\_10N following a quadratic relationship (R^2 = 0.62). The enhanced k\_660 (blue dots in Fig. 7), showed a wind-speed dependency of higher order (cubic) and R^2 = 0.57."

---

## Author Comment (AC2)

MS No.: bg-2022-82
Authors: Lucía Gutiérrez-Loza et al.
https://doi.org/10.5194/bg-2022-82

**Authors' reply to Bernd Jähne**

Dear Bernd Jähne,

We would like to thank you for taking the time to comment on this manuscript. Please find below our replies to your comments, which we have tried to address in the best way possible.

*The response to each comment is written in blue italics, while the changes made in the revised manuscript are in red.*

1. Empirical correlation instead of analysis of mechanisms

The authors claim in their paper to analyze the physical mechanisms controlling air-sea gas transfer beyond wind speed using a nine year record of continuous eddy covariance measurements taken at the Östergarnsholm tower. Such a long time series is, of course, a treasure, but the title of the paper is misleading in my view. The paper does just list possible other mechanisms than wind (or more precisely wind stress at the ocean surface) as the main driving force, but does not discuss at all any possible conceptual model to be checked by the data. All they do are empirical correlations with some other parameters. In addition, even an observed correlation does not mean that another driving force is found, because the investigated parameter might be correlated to another, which is the real driving force.

Therefore a more appropriate title would be "Empirical correlations of the gas transfer velocity with other parameters than wind speed" or a similar wording.

*In the revise manuscript, a different approach was used (brief discussion comes as a reply to the comments below). This approach allowed us to identify conditions at which the gas transfer velocity showed higher values than those predicted by the k_660-wind relationship found in the current study, as well as by other wind-based parametrizations used as reference (W14 and Mc01). By identifying these conditions, it was possible to explain part of the variability observed at high and low wind speed conditions. We, therefore, considered that a more suitable title for the manuscript is "On physical mechanisms enhancing air-sea CO2 fluxes".*

*The **title** of the manuscript was modified to "**On physical mechanisms enhancing air-sea CO2 exchange**".*

2. The statistical analysis is not convincing

- The authors use a multistep scheme to select only reliable measurements. But they do not specify which fraction is remaining. It is important to know to which extent specific conditions are excluded from the analysis. If the exclusion were large, this may severely bias any averaging and correlations.

*The authors agree with this comment. The data selection process implied that a very large percentage of the data was discarded from the final data set used in the analysis. Additional information was included in the manuscript to clarify this fact (see below).*

*The final data set consisted of 18.7% of the initial FCO2 data and 15% of the k660 data, with respect to the total amount of data available for the open-sea sector.*

*Furthermore, we would like to point out that, even if the exclusion of data was indeed large, the distribution of the data shows similar patterns between the initial data set (before quality control) and the final data set (after quality control). Indicating that the statistical representation of the data is adequate and only small biases are expected. Some figures are included here as an example:*

[Figure]

*Figure CC1.1. Probability density function of U10N (upper left), ΔpCO2 (upper right), Hs (lower left), and stability (z/L, lower right) for all data available for the open sea sector (yellow line) and for the processed (quality controlled) data used in the analysis (blue line).*

*A paragraph including some general numbers indicating the size of the initial data set (for the open-sea sector) and the remaining amount of data after quality control was included at the end of the **Data Processing** section (**2.2**).*

***Appendix B** was included. In this section, a more detailed description of the relative importance of every QC criterion (i.e. the effect of each criterion on the total amount of data) was included (**Table B1**).*

- For the averages shown in Fig. 5 all, also negative gas transfer velocities have been used, but not in the following more detailed analysis. This is inconsistent. Either you rely data or not at all.

*We agree with this comment, removing the negative k values was inconsistent. We have initially thought that these negative k values would introduce some unrealistic results as we could not find a feasible explanation for them, even if they had fulfilled all quality control steps. However, we have reconsidered this decision and have included the negative k values in the revised version as part of the analysis. Thus, avoiding biases in the data and the subsequent analysis.*

*All data fulfilling the quality control procedures were considered in the analysis, including the negative k_660 values. The corresponding modifications were made throughout the manuscript. In particular, modification were made to the data processing **section 2.2** and in the results from **section 3.2** and onwards.*

- The concept of the residual gas transfer velocity and the correlation with a single other parameter seems not to be a good choice. A theoretically more sound approach would be to use a multi-parameter space approach with all possible parameters and then perform a principal component analysis or similar approach, as it is standard in pattern recognition and classification. Then it would be possible to identify the most important parameters and the degree of correlation between the parameters quantitatively and whether the results are statistically relevant. The approach of the authors is just qualitatively.

*Based on this, and other comments from the reviewers, we have reconsidered the use of the residual gas transfer velocity (k_r). Instead, a different approach using a normalized gas transfer velocity was used. Such normalized gas transfer velocity defined as k_660/k_wind, where k_wind was obtained as the cubic (best) fit to the bin-averaged k_660 data of the current study. This approach allowed us to quantify the difference between k_660 from the expected wind-speed relationship. Based on these values, conditions leading to significant deviations of k_660 from k_wind, were identified and further analyzed. We consider this method to be, statistically, more robust than the previous approach which, as suggested, was based on*

*qualitative relations. A description of the changes made (including figures) can be found in the response to the comments from Reviewer 1 and 2.*

**Data processing section (2.2**). *Information regarding k_r was removed and a short paragraph describing the calculations of the normalized gas transfer velocity was included.*

*Corresponding changes in the **Results** section, from **sub-section 3.2** and onwards, as well as in the **Discussion.***

*Furthermore, a preliminary attempt of using a multivariate approach (Partial Least Squares, (PLS) analysis) was considered to analyze the data, as suggested. We considered this method, similar to PCA, to be suitable for this particular application as it can handle data that are non-normally distributed, as well as accounting for the effect of cross-correlated predictor variables. The results of the PLS analysis suggest that, for high-wind speed conditions, it is possible to explain up to 90% of the variability in k_660 by using five components (up from 65.2% using only one component). At low wind speed conditions, on the contrary, increasing the number of components in the analysis only improves the model by 11.3% (from 61.6% with one component to 73.2% with five components). In both, high and low wind-speed conditions, the model with five components was considered to be the best base on the RMSEP and explained variability. These results, suggest that even if a strong correlation existed between predictor variables (e.g. wind speed and significant wave height) the additional contribution of parameters, other than wind speed, to the explained variability is relevant. In particular, under high wind speed conditions. Accounting for the effect of additional control mechanisms, seems therefore relevant to further explain the variability of k_660.*

*Even when PLS (or other statistical models) can provide insights about the relationship between k_660 and other parameters, the characteristics of the data set such as the presence of large gaps in the data, non-normal behavior, cross-correlation of predictor variables, etc. make it hard to interpret the obtained results. For this reason, such statistical analysis was not included in the revised manuscript.*

3. Claimed effect of sea spray in enhancing air-sea CO2 fluxes questionable

The authors claim that "sea spray can enhance the transport of CO2 across the interface, as it does with heat and moisture". This is an oversimplification, because the transport of heat and water vapor is air-side controlled, while the transport of CO2 is water-side controlled. Therefore a much more detailed analysis is required. Droplets may partly be a dead surface for gas transfer in the same way as those bubbles, which come in a much shorter time scale into equilibrium with the surrounding water than their life time. However, the influence of solubility is inverse. Droplets are efficient for highly soluble tracers.

*Yes, we suggest that sea spray can enhance the transport of CO_2, but only under particular conditions. Here, we are not challenging the well-stablished theory about the water-side controlled transport of CO2. However, we acknowledge (as other studies have) that atmospheric processes might also be relevant for air-sea CO_2 gas exchange under particular conditions (e.g. effect of atmospheric stability). We offered, based on the analysis presented here, a plausible explanation for the higher transport observed under such conditions suggesting the potential effect of sea spray on FCO_2. This is not to say that sea spray does not occur under*

*other conditions but, as expected from the water-side controlled transport perspective, it is not expected to affect FCO_2.*

*The approach used in the revised manuscript allowed us to identify (in a clearer way) these conditions, and to show that such conditions were consistently linked with higher values of k_660 than those predicted by wind-speed relationships (Fig. 7 of the revised manuscript). However, we recognize that further investigation is necessary to understand this transport mechanism, and to quantify its impact on the regional and global fluxes.*

*Further investigation of the sea spray as a control mechanisms of air-sea CO_2 exchange is suggested in the **Discussion** and **Conclusions.***

---

## Author Comment (AC3)

MS No.: bg-2022-82
Authors: Lucía Gutiérrez-Loza et al.
https://doi.org/10.5194/bg-2022-82

**Authors' reply to Anonymous Referee #2**

Dear reviewer,

We would like to thank you for reviewing this manuscript. We have addressed your comments and provide with the answers below. We consider that the manuscript has improved significantly based on you feedback, and we are grateful to you for that.

*The response to each comment is written in blue italics, while the changes made in the revised manuscript are in red.*

This manuscript by Gutiérrez-Loza et al. presents a new dataset of eddy correlation air-sea CO2 flux measurements that is in itself a valuable contribution to the field. Analysis of the dataset has the potential to improve our understanding of gas exchange processes and the analysis done here does reveal some new insights particularly regarding the importance of water-side controls on the gas exchange rate and the influence of processes that cannot be parameterised as a function of wind speed. It is generally well written, interesting and easy to follow. I encourage the authors to work further on the dataset to see it through to publication.

However, there are a few major issues that would be essential to address first:

**Major issues:**

- How representative are the results here in a global context? The title leads one to expect that these will be universally applicable insights, but this does not seem to be the case. Indeed the most confidence the authors were able to express in the wider applicability of their results was 'the results presented here are most probably relevant for other marginal seas and coastal areas' (line 375, emphasis mine). Line 314 also suggests this work may be only relevant to the Baltic. The title should be further qualified '… in the Baltic Sea' or similar unless the authors can be sure that their results are more widely applicable.

*The approach used in the revised manuscript (see replies further down in this document) proved to be a more suitable, and statistically robust, method for the analysis of k_660. The results from this approach were used to identify conditions enhancing the gas exchange. We further linked such conditions to the potential effect of sea spray (under high wind speeds) and water-side convection (under low wind speed conditions). We consider these two mechanisms to be potentially relevant in other regions and spatio-temporal scales. Detailed analysis of the wider*

*applicability of these results is beyond the scope of this work. However, the potential relevance of these mechanisms is discussed in the revised manuscript.*

*The **title** of the manuscript was changed to "**On physical mechanisms enhancing air-sea CO2 exchange**".*

*Discussion about the potential relevance of sea spray and water-side convection in other regions was included in the **Discussion** section.*

- One of the key motivations for this study is reducing uncertainties in gas exchange calculations (e.g. lines 22 – 24) yet there is no meaningful uncertainty analysis of the results of this study. I could not even see uncertainty estimates for the raw measurements that underlie the new dataset being presented and there was no propagation of uncertainty through to the final results. This is essential especially if the results are to be compared with previous work or other approaches, else you cannot be sure if the results are actually consistent or not.

*In this sentence (lines 22-24) we acknowledged the existence (and relevance) of the uncertainties associated to the air-sea CO_2 fluxes at a global scale. Furthermore, we stated that a large proportion of these uncertainties exist due to the "incomplete understanding of the spatio-temporal variability in the controlling mechanisms". The intention of this sentence is not to set up the exact focus of current study, but rather to put into context why more process-oriented studies are necessary, and the implications in the global context.*

*The focus of our study is to capture the temporal variability of k_660 and other processes involved in the exchange. We assessed this by presenting long records of high-frequency data, thus, capturing both the short- and long-term (several years) variability of FCO_2 and k_660. In the revised manuscript, we presented a more statistically robust analysis of the data and used it to identify mechanisms that can potentially cause large deviations on k_660.*

*A subsequent sentence was included in the first paragraph of the introduction. This with the aim of highlighting the relevance of resolving the different mechanisms involved in the gas exchange:*

*"*However, large uncertainties are still associated with the air--sea CO_2 flux estimates, mainly due to the incomplete understanding of the spatio-temporal variability in the controlling mechanisms. **Resolving the effect of these mechanisms at the relevant temporal and spatial scales is essential to constrain the oceanic contribution in the global carbon balance**.*"*

- Many of the relationships described in section 3.2 and its subsections were not convincing based on the figures. For example on line 225 'kr showed a clear relationship with significant wave height … (Fig 6a)' but if we look at Fig 6a, then I see more clearly the colours getting 'higher' in more vertical bands towards the right (ie correlating with U10) rather than vertical bands towards the top. Same applies for line 239 comment about mixed layer depth. But in general I think that the format of Figs 6-10 makes it very hard to see the correlations described anyway: you are trying to eyeball the angle at which changes in colours occur and there are so many datapoints that they all block each other (e.g. fig 8b has a section in the middle that

looks all blue i.e. low values, but with hints of higher pink points that can just be seen around the edges – very hard to interpret). My suggestion would be to replot these figures with kr on the x-axis and the variable of interest (currently the colours) on the y-axis. The points could then be coloured by U10 or something else. This would be a much more clear and convincing way to see correlations. Furthermore, related to point (2) above, there needs to be statistics for the correlations that you report.

*We thank Reviewer 2 for this comment. Based on this and other comments received on the original submission, we decided to reframe the way the analysis was made and, as suggested here, the way the data is presented in the revised manuscript. Firstly, the data is no longer analyzed using the residual gas transfer velocity (k_r). Instead, we use a normalized gas transfer velocity for each 30-min average, defined as k_660/k_wind, where k_wind is the best fit to the bin-averaged k_660 values of the current study. Secondly, the normalized gas transfer velocity was plotted (for each wind speed regime) as a function of each of the variables of interest (i.e. ΔpCO2, MLD, Hs, etc.). We consider that, given the complex relationship between the wind speed and the other parameters evaluated in this study, it would be hard to find meaningful correlations between k_660 and those parameters. However, based on the analysis of the normalized gas transfer velocity, it was possible to identify a set of conditions leading to deviations of k_660 from k_wind. The analysis was carried out using boxplots which provide statistical summaries of the data. Subsequent analysis was then made based on the behavior of k_660 under the enhanced conditions found.*

*Figures 6 to 10 were substituted by the following figures:*

[Figure]

**Figure 6.** Normalized gas transfer velocity ($k_{660}/k_{wind}$) under high wind speed conditions ($U_{10N} > 8\,\mathrm{m\,s^{-1}}$) as a function of a) $\Delta pCO_2$, b) atmospheric stability ($z/L$), c) mixed layer depth (MLD), d) relative humidity (RH), e) significant wave height ($H_s$), and f) total enthalpy flux ($F_k$). The crosses represent the individual half-hourly values. The boxplots give a statistical summary for equidensity bins defined based on the distribution of $k_{660}/k_{wind}$ as a function of each of the parameters (see Appendix A). The median, first, and third quartiles are represented in each box; the whiskers represent the minimum and maximum values, and the black dots represent the outliers; the notches highlighted in pink indicate the median's 95 % confidence interval. The open circles linked with a dashed line indicate the bin means, and the horizontal red line indicates $k_{660}/k_{wind} = 1$.

[Figure]

**Figure 7.** Gas transfer velocity for $CO_2$ (adjusted to a Schmidt number of 660) as a function of the 10 m neutral wind speed. The dots represent the half-hourly values of $k_{660}$. The blue dots represent $k_{660}$ under enhanced conditions (see text for details), while the blue dots with a black edge indicate cases where $H_s > 1.5$ m. The black line represents the best fit (quadratic) to the data excluding the enhanced cases (only gray dots), while the pink line is the best fit to the enhanced data (only blue dots). For reference, a quadratic (Wanninkhof, 2014) and cubic (McGillis et al., 2001) wind-based parametrizations were included. The wind speed regimes are separated by vertical dashed lines.

[Figure]

**Figure 8.** Normalized gas transfer velocity ($k_{660}/k_{wind}$) under low wind speed conditions ($U_{10N} \leq 6\,\mathrm{m\,s^{-1}}$) as a function of a) $\Delta pCO_2$, b) atmospheric stability ($z/L$), c) water-side convective scale ($w*$) under unstable atmospheric conditions, and d) enthalpy flux ($F_k$). The crosses represent the individual half-hourly values. The boxplots give a statistical summary for equidensity bins defined based on the distribution of $k_{660}/k_{wind}$ as a function of each of the parameters (see Appendix A). The median, first, and third quartiles are represented in each box; the whiskers represent the minimum and maximum values, and the black dots represent the outliers; the notches highlighted in yellow indicate the median's 95 % confidence interval. The open circles linked with a dashed line indicate the bin means, and the horizontal red line indicates $k_{660}/k_{wind} = 1$.

[Figure]

**Figure 9.** Gas transfer velocity for $CO_2$ (adjusted to a Schmidt number of 660) as a function of the 10 m neutral wind speed during a) winter and b) summer. The dots represent the half-hourly values of $k_{660}$. The color represents the water-side convective scale ($w*$) for data under unstable atmospheric conditions, calculated according to Rutgersson and Smedman (2010). The wind speed regimes are separated by a vertical dashed line.

[Figure]

**Figure 10.** Boxplots of the gas transfer velocity for $CO_2$ (adjusted to a Schmidt number of 660) during unstable atmospheric conditions as a function of the 10 m neutral wind speed during summer (pink) and winter (blue). The median, first, and third quartiles are represented in each box; the whiskers represent the minimum and maximum values, and the circles and crosses represent the outliers; the notches highlighted in color indicate the median's 95 % confidence interval. The wind speed regimes are separated by a vertical dashed line.

- Many parts of the dataset are excluded and the impact of this on the results and their wider applicability is not much discussed. We have low wind speeds on line 139, low fluxes on 154, high humidity on line 157, stratified conditions on line 169, and unexplained low k660 values on line 174. Maybe it's valid to not include these in the analysis, but we really need an accompanying robust discussion of how often those conditions occur in the real world and what that means for the gas exchange rate.

*We agree that further clarity and transparency was necessary when discussing the quality control and the amount of data rejected, as well as the implications of rejecting such data from the analysis of the gas transfer velocity. This issue was tackled by including a section (**Appendix B**) describing the relative importance of each quality control criterion.*

***Appendix B** was included where a more detailed description of the relative importance of every QC criterion (i.e. the effect of each criterion on the total amount of data) is presented and brief discussion about the final size of the data set.*

**Table B1.** Percentage of data that successfully fulfill each individual quality control criterion. The percentages are relative to the total recorded amount of data (100% = 125,001 for $FCO_2$ and 100% = 66,475 for $k_{660}$).

| Quality control criteria | $FCO_2$ data (%) | $k_{660}$ data (%) |
|---|---|---|
| $U_{min} = 2\,\mathrm{m\,s^{-1}}$ | 95.1 | 94.5 |
| Signal quality ($\sigma^2_{RSSI} < 0.001$) | 36.7 | 41.7 |
| Turbulence level ($\sigma^2_w > 1e^{-6}\,\mathrm{m^2\,s^{-2}}$) | 99.4 | 99.4 |
| Remove outliers | 80.0 | 81.7 |
| $|FCO_2|_{min} = 0.05\,\mathrm{\mu mol\,m^{-2}\,s^{-1}}$ | 51.4 | 48.4 |
| $|\Delta pCO_2|_{min} = 50\,\mathrm{\mu atm}$ | N/A | 41.3 |
| RH$< 95\,\%$ | 89.2 | 90.3 |
| Open-sea sector ($80° < $WD$ < 160°$) | 14.9 | 13.5 |
| $|\Delta T_w|_{max} = 1\,°\mathrm{C}$ | N/A | 77.6 |

N/A = Not applicable. The corresponding criterion has no impact on the resulting amount of data.

*Furthermore, an initial evaluation of the probability distributions of several parameters included in the analysis showed that the initial data set (before quality control) and the final data set (after quality control) had similar patterns. Thus, indicating that the QC-ed data set used in the k_660 analysis was representative of the local conditions and biases due to the exclusion of data are expected to be small, if any. This analysis was not included in the manuscript but figures can be found in the response to the comments of Bernd Jähne (community comment 1).*

*Brief discussion of the implication of removing data under certain criteria (e.g. U10N<2 m/s), was included as part of the **Discussion** section.*

- Despite high variability and physical/biogeochemical heterogeneity being an important motivator of the study, some key properties were assumed to be uniform (salinity on line 116, somewhat cryptic 'biogeochemical water properties' on line 101). This may be fine but this assumption is not critically assessed. One should be able to quantify a maximum effect size for how important ignoring this variability could be.

*We acknowledge the fact that there is a significant variability in the biogeochemical properties, however, some assumptions had to be made during the analysis due to practical reasons. We believe, however, that this assumptions are well founded. For instance, the use of a constant salinity value for the solubility calculations. As stated in the manuscript, salinity values range between 6.5 and 7.5 PSU in the area of study, which in turn have a small impact on the $CO_2$ solubility when compared with the effects of temperature (see Weiss, 1974), which also presents a much larger variability in the current site. Furthermore, the final effect of salinity on the gas transfer velocity calculations is of the order of $1x10^{-2}$ cm/h per PSU (based on an initial sensitivity analysis). Such effect was considered to be negligible given the low salinity range.*

*References to **Wesslander et al., 2010** and **Rutgersson et al., 2020** were included. In that study, long records of sea surface salinity in the central Baltic Sea show its limited variability.*

*On the other hand, the assumption about the homogeneous biogeochemical water properties is based on the wind-direction classification discussed in Rutgersson et al, 2020. This*

*consideration was made in order to be able to assume that the seawater pCO_2 measurements (recorded at a single location) were representative of the entire open-sea sector; furthermore, that the fluxes measured at the tower can be associated to these seawater measurements. In **section 3.2.2** we stated that "the strong stratification, relatively weak ΔpCO_2, and the possibility of strong heterogeneity in terms of the biogeochemical properties might hinder our capacity to calculate k_660 from pCO_2w and FCO_2". We, therefore, suggest that "the interpretation of these data should be taken with some caution".*

*A detailed assessment about the wind-direction categories, including the analysis of the water properties in the region is presented in Rutgersson et al., 2020. For such analysis, observations of temperature, salinity, pCO2, dissolved oxygen, pH, chlorophyll-a, and nutrients in the vicinity of Östergarnsholm were taken into account.*

*In **Section 2.1.1**:*

"Furthermore, the biogeochemical water properties **and the hydrographical features** were assumed to be **spatially** homogeneous **along** this sector (Rutgersson et al., 2008, **2020**), **ensuring that the water-side measurements were representative of the flux footprint of the tower.**"

**Other minor comments:**

46 how deep is 'the upper layer of the ocean'

*The upper layer of the ocean can be considered to be the layer that is adjacent to the air-sea interface and that is affected by processes at the ocean surface (i.e. meteorological conditions, wave field, radiation, etc.). However, there is no universal definition of what the depth of this layer is, being strongly dependent on the local and regional conditions. In general, the depth of the upper layer of the ocean could be considered to be from a few meters to several tens of meters (e.g. Soloviev and Lukas, 2013, Moum and Smyth, 2019).*

*The sentence was modified to:*

"At moderately high wind speeds, above 8-10 m s-1, the upper layer of the ocean is generally well mixed **(from the surface up to several tens of meters depth)**."

Fig 1 caption 'see text for details' of open sea sector – please give section number and repeat values here

*Changes were made accordingly.*

*Caption in **Figure 1** was modified to:*

"(a) Map of the Baltic Sea; the red cross in the central Baltic Sea indicates the location of the Östergarnsholm station. (b) Map of the Östergarnsholm station ca 4 km off from the Gotland Island; the red dot indicates the location of the tower, the blue cross is the location of the mooring with water-side instrumentation (Sect. 2.1.2), and the shaded blue area is the so-called "open-sea" sector **with wind directions from 80°< WD < 160°** (see **Sect. 2.1.1** for details)."

Fig 2 Very unclear to me what this figure shows. What are X and Y on the axes? What does 'footprint' mean?

*The flux footprint, f(x,y), is a transfer function used to relate the sources or sinks of scalars at the surface with the measurements made at a specific height (Kljun et al. 2015). The footprint represents the contribution per unit area of each unit (i.e. $m^2$) source or sink ($Q_u(x,y)$) to the total flux measured ($F_c(0,0,z)$), and the units are expressed in $m^{-2}$:*

$$f(x,y) = \frac{F_c(0,0,z)}{Q_u(x,y)} = \left[\frac{\mu mol\ m^{-2}s^{-1}}{\mu mol\ s^{-1}}\right] = [m^{-2}]$$

*In Figure 2, the average footprint of the fluxes at the tower is presented for different atmospheric stability conditions. There, X and Y are length scales in meters using the reference point (0,0) as the tower where the measurements are made.*

*Some clarifications were made in the manuscript, however, an in-depth description of this concept is beyond the scope of this work.*

*A brief description of the concept of footprint was included in **Section 2.1.1 (Atmospheric data)**:*

"The flux footprint is a function used to characterize the contributions of the sources and sinks per unit area to the total flux measured at a certain point. Based on this mathematical concept, it is possible to associate the fluxes measured at a specific height with the surface exchange of any scalar (Kormann and Meixner, 2001)"

*A more concrete description of the content of **Figure 2** is now given:*

"Figure 2 shows the spatial distribution of the flux contributions (in $m^{-2}$) for different atmospheric stability conditions"

*Additional information about the color scale in the figure is given in the caption of **Figure 2**:*

"Average footprint distribution for (a) unstable, (b) neutral, and (c) stable atmospheric conditions. The green cross indicates the position of the tower and the contours represent the percentage of source area from 10-80%. **The flux footprint (in color) shows the spatial distribution of the contributions per unit area to the total FCO2 (in $m^{-2}$)**. The footprint was calculated using the model developed by Kljun et al. (2015) using all data available for the open-sea sector between mid-2013 and 2020"

161 'in these region' => 'in this region'

*The error was corrected.*

175 'more detail analysis' => 'more detailed analysis'

*The sentence has been removed from the text. In the revised manuscript, the negative k values were included in the analysis, thus, the sentence was no longer needed.*

183 mixed up > and < symbols for intermediate conditions

*Thanks for pointing this out. The error was corrected.*

Section 3.1 worth pointing out that the seasonal cycle of pCO2w looks to be biologically controlled rather than temperature controlled in this part of the world – any impact on wider applicability? See e.g. analysis of Takahashi et al (2009, DSR2)

*In addition to the analysis presented by Takahashi et al., 2009, there have been studies suggesting that the seasonal cycle of pCO2w is, to a large extent, controlled by biological activity in the Baltic Sea (e.g. Helmuth and Schneider, 1999, Wesslander et al., 2010). The seasonality occurs in both, the local biological activity and the transport of organic matter. A detailed analysis of these processes is beyond the scope of this study. However, we agree with Reviewer 2 and we, therefore, mention the relevance of the biological activity on the seasonal cycle of pCO_2w.*

*The following sentence was included in **Section 3.1**:*

"The seasonality in pCO_2w in the Baltic Sea has been recognized to be strongly modulated by the biological activity (Helmuth and Schneider, 1999, Wesslander et al., 2010)"

Fig 3 State which method of calculating air-sea CO2 fluxes is used here

*Text was modified accordingly.*

*Caption in **Figure 3** was modified to:*

"Annual cycle of (a) CO2 partial pressure (pCO2) in the seawater and in the atmosphere, and (b) air–sea CO2 fluxes **from eddy covariance**. The dots represent the half-hourly values while the solid lines show the monthly averages."

228 please explain briefly why the wave age suggests local generation of waves (for the non-expert)

*The wave age is a measure of the effect of the wind over the wave field. Therefore, when the ratio between Cp and U10N is small, there is an indication that the effect of the wind forcing on a specific wave group is large, thus, generating wave growth. On the contrary, when this ratio is large, the wave field is developed and the forcing of the wind has little effect on the waves.*

*A brief explanation about why the observed wave age values suggests locally-generated waves was included:*

"While **the small values of** the wave age ($C_p/U_{10N}$, Fig. 6c) **suggest wave growth caused by the forcing of the wind over wave field. Thus, indicating** locally generated waves (i.e. wind sea) at these wind speed conditions."

Figs 4 & 5 suggest to make points smaller and semitransparent so that structure within the big grey overlapping blob can be seen

*Thank you for the comment. We agree that the data can be tightly clustered and overlapping in some areas, making it difficult to visualize individual data points. Changes were made in Figure 5 to avoid this issue and improve the visualization of the data as much as possible. However, for Figure 4, the amount of data is significantly higher (less data are removed during quality control for these variables in comparison with k660). Visualization of individual data points is hard with such amounts of data, thus, we considered that showing the figure as is still give a good-enough understanding of the general behavior of the data.*

*Changes in **Figure 5** were made. The dots were made smaller and with white edges to make it easier to see when dots are overlapping. Additional color shading was included to indicate the data density. The legend was modified accordingly:*

[Figure]

**Figure 5.** Gas transfer velocity for $CO_2$ (normalized to a Schmidt number of 660) as a function of the 10 m neutral wind speed. The grey dots represent the half-hourly values of $k_{660}$ for the nine-year period from 2013 to 2021. The black dots and bars, represent the $k_{660}$ mean values and standard deviations, respectively, calculated for equi-density bins based on the wind speed percentiles; the best fit to the means is shown as the pink dotted line. For reference, a quadratic (Wanninkhof, 2014) and cubic (McGillis et al., 2001) wind-based parametrizations were included.The colors in the shaded area represent the data density (in counts) with a grid bin size of $1\,\mathrm{m\,s^{-1}}$ by $10\,\mathrm{cm\,h^{-1}}$.

300 there are studies that compare these other parameterisations, have you looked at those to put your comparison in more context?

*The quadratic (W14) and a cubic (Mc01) parametrizations were included in the manuscript as a reference. However, further analysis of these, or other, wind-based parametrizations was beyond the scope of this study.*

305-306 not convinced that this long-term average being correct but short term was really 'shown' here. Needs statistics and more rigorous definitions (what is long term? How much uncertainty is there by ignoring the water side effects?)

*We have decided to keep the wind-based as a (visual) reference, thus, no further comparison was made between our data and these parameterization. With this in mind, we decided to remove the statements in which we suggested that existing wind-based parameterization might be good representations of the gas exchange in the study region. Instead, we developed the discussion around the k_wind, which was obtained from data presented in this study. Furthermore, we avoided the use of long-term (in the context referred here) and instead use the words "average" or "bin-averaged".*

*We would like to point out that we are not "ignoring the water side effects" in the analysis by using EC data. On the contrary, by using EC we are directly measuring the total flux, thus, accounting for all the processes involved in the transport (even if these are not measured individually). We apologize if we are misunderstanding this question.*

394 define 'adequately' (adequate for what purpose?)

*This particular phrase was removed. We agree that further comparison between our data and existing wind-based parameterizations would be necessary in order to find out how adequate these parametrizations are for this study site.*

References:

Thomas, H., & Schneider, B. (1999). The seasonal cycle of carbon dioxide in Baltic Sea surface waters. *Journal of Marine Systems*, *22*(1), 53-67.

Kljun, N., Calanca, P., Rotach, M., and Schmid, H. P. (2015). A simple two-dimensional parameterisation for Flux Footprint Prediction (FFP), *Geoscientific Model Development*, 8, 3695–3713

Kormann, R., and Meixner, F. X. (2001). An analytical footprint model for non-neutral stratification. *Boundary-Layer Meteorology*, *99*(2), 207-224.

Moum J.N., and Smyth W.D. (2019). Upper Ocean Mixing. In Cochran, J. Kirk; Bokuniewicz, J. Henry; Yager, L. Patricia (Eds.) *Encyclopedia of Ocean Sciences*, 3rd Edition. vol. 1, pp. 71-79, Elsevier. ISBN: 978-0-12-813081-0

Soloviev, A. and Lukas, R. (2013). The near-surface layer of the ocean: structure, dynamics and applications, Springer Science & Business Media, vol. 48.

Weiss, R. (1974). Carbon dioxide in water and seawater: the solubility of a non-ideal gas. *Marine Chemistry*, 2, 203-215.

Wesslander, K., Omstedt, A., and Schneider, B. (2010). Inter-annual and seasonal variations in the air–sea CO2 balance in the central Baltic Sea and the Kattegat. *Continental Shelf Research*, *30*(14), 1511-1521.

---

## Author Response (AR1)

MS No.: bg-2022-82
Authors: Lucía Gutiérrez-Loza et al.
https://doi.org/10.5194/bg-2022-82

**Authors' reply to Associate Editor Peter Landschützer**

Dear Peter Landschützer,

We want to thank you for giving us the opportunity to submit the revised version of our manuscript "On physical mechanisms enhancing air-sea $CO_2$ gas exchange". We appreciate your time, and the time of the reviewers, whose comments provided invaluable contributions for the improvement of this manuscript.

Thank you for your consideration of this manuscript.

Sincerely,
Lucía Gutiérrez-Loza et al.

---

## Author Response (AR2)

MS No.: bg-2022-82
Authors: Lucía Gutiérrez-Loza et al.
https://doi.org/10.5194/bg-2022-82

**Authors' reply to Associate Editor Peter Landschützer**

Dear Peter Landschützer,

We want to thank you for taking the time to review our manuscript again and giving us the opportunity to further improve it. Based on your comments, we have included a short discussion regarding the uncertainties associated to the flux measurements and the implications for the gas transfer velocity calculations. This discussion if followed by the description of the two (additional) major limitations we encountered during the analysis.

Sincerely,
Lucía Gutiérrez-Loza et al.